# Distribution of microbial carrageenan foraging pathways reveals a widespread latent trait within the ruminant intestinal microbiome

Jeffrey P. Tingley[1,2], Thea O. Andersen [3,4], Liam G. Mihalynuk[5], Xiaohui Xing [1], Kristin E. Low[1], Douglas P. Whiteside [6,7], Ianina Altshuler [3,8], Nic Jujihara[1], Anna Y. Shearer[1,2], Leeann Klassen[1], Spencer Serin[9], Edgar Smith[10], Greta Reintjes [11], Trushar R. Patel [2,12,13], Alisdair B. Boraston [5], Live H. Hagen [4], Phillip B. Pope [3,4,14] & D. Wade Abbott [1,2] ✉

Seaweeds represent a promising source of sustainable, alternative feeds for livestock. Despite their increasing popularity in agriculture, the dietary fate of seaweed polysaccharides, such as carrageenan, is unknown. Here, we apply functional microbiome analyses of ruminant gastrointestinal tract microbiomes to discover catabolic enzymes specific for carrageenan digestion from the red seaweed *Mazzaella japonica*. *M. japonica* preferentially increased *Bacteroides* abundance within the feces over the rumen, and bacterial isolates have the capacity to use carrageenans as a sole carbon source. We identify carrageenan-active polysaccharide utilization loci (CarPULs) and characterize recombinant GH16 subfamily 17 carrageenases, informing previously uncharacterized substrate specificities for the subfamily, and providing insights into pathway specialization of divergent CarPULs. Selective enrichment and metagenomic mining reveals that carrageenan catabolism is widespread among geographically and taxonomically distinct ruminants, suggesting it is a latent trait widely distributed in the Order Artiodactyla and carried within their microbiomes as part of the microbial "dark matter". These pathways are structurally distinct from those found in marine bacteria, highlighting a complex and ancient evolutionary history of CarPULs in ruminant microbiomes.

With growing global concerns about greenhouse gas emissions, the need for sustainable agricultural practices continues to escalate. Enteric methane is a by-product of ruminant fermentation[1–3] and exerts a stronger warming effect than carbon dioxide despite possessing a shorter half-life[4]. Some seaweeds, such as *Asparagopsis taxiformis*, can act as inhibitors of methane emissions when provided as a feed additive to cattle[5–7]. Seaweed has been utilized for millennia to supplement cattle and sheep diets in herds raised near coastlines and represents a viable but underutilized alternative feed source[8,9].

Ruminants rely on established syntrophic relationships within their gastrointestinal tract (GIT) microbial community to digest and extract energy from fiber in their diets. Many of these microorganisms are endowed with extensive inventories of carbohydrate-active enzymes (CAZymes) that modify and depolymerize chemically and

structurally complex polysaccharides[10,11]. These include "keystone" endo-acting enzymes[12–14], which are surface-associated and catalyze initial polysaccharide fragmentation; and sulfatases[15], which remove obstinate sulfates preventing saccharification. Among the dominant phyla residing in ruminant GITs, gram-negative *Bacteroidota* have been intensively studied as saccharolytic generalists known for their metabolic plasticity[15]. *Bacteroidota* commonly possess extensive CAZyme collections, organized into independently regulated polysaccharide utilization loci (PULs), to catabolize discrete polysaccharide substrates[10,16]. PUL inventories within the genome determine the diversity of substrates consumed by individual strains[17], and therefore, acquisition or loss of PULs are important considerations for determining responses to shifts in diet or exposure to exotic polysaccharides, such as those found in seaweeds[18–20]. Elucidating the processes driving PUL acquisition and their timelines can be difficult. For example, the xyloglucan PUL found in *Bacteroides* spp. lies between ancestral genes shared by many strains with no evidence of lateral transfer[21]. Other PULs display residual genetic signatures consistent with horizontal gene transfer events. These include yeast mannan PULs found in western human GIT microbiomes[22], or dietary seaweed polysaccharide PULs, found predominantly in east Asian human GIT microbiomes[20], which can be specific for porphyran[18], agarose[19], or carrageenan[20].

Dietary selection and extinction pressures can drastically influence which metabolic capabilities are retained within the microbiome[23]. *Bacteroidota* has the capacity to digest multiple complex polysaccharides, yet how long strains and PULs persist when their cognate substrates are in limited amounts or absent is not known. Previously, we have shown that alginate PULs are functionally conserved in two different ruminant species that were naïve to dietary seaweed and raised on two different continents[24]. We have proposed that these "latent traits" maintained in GIT microbiomes represent a common nutritional strategy exploited by ruminants and other herbivores to adapt rapidly to dietary changes. Therefore, understanding how bacterial strains persist and unused genetic repositories are maintained within the genetic "dark matter"[25] of the microbiome warrants further study.

To understand the process of polysaccharide consumption by host microorganisms, it is important to know the chemical structure of the polysaccharide[26], which microorganisms and metabolic pathways respond to the substrate[27], and the function of enzymes within associated catabolic pathways[28]. Further, disentangling the evolutionary origins of PULs can aid in understanding the timelines of pathway acquisition and the processes driving their functional divergence.

Here we elucidate how *M. japonica* cell wall carrageenans are consumed by *Bacteroidota* that were enriched in the fecal microbiomes of cattle. Selective enrichment and primary sequence BLAST analysis reveals diverse carrageenan PUL (CarPUL) structures with functional specialization within ruminant members of Order Artiodactyla. The implications for how naïve cattle and other herbivorous artiodactyls adapt to dietary carrageenan, and the origin of latent traits responsible for their digestion, are considered.

## Results

### Degradation of carrageenans within cattle is facilitated within the lower GIT by Bacteroides harbouring carrageenan-degrading PULs

Two cattle trials were conducted using *M. japonica* supplemented in diets with distinct animals. In the first trial, *M. japonica* was provided to cattle *ad libitum* on pasture; in the second, cattle were fed 5% *M. japonica* as a silage in covered pens. Using 16S rRNA gene sequencing, few compositional differences were observed within the rumen microbiome of animals in the *ad libitum* trial; however, there was a minor, yet significant, increase in *Bacteroidaceae* observed in the silage trial (Supplementary Fig. 1A, B). In contrast, the fecal communities in both the *ad libitum* and silage trials displayed a significant increase in *Bacteroidaceae* when the animals were fed seaweed (Fig. 1A). This is highlighted by an increase in *Bacteroides*, at the apparent expense of the uncultured *Bacillota* lineage UCG-010 (Fig. 1B). Overall, *M. japonica* supplementation selectively influences the fecal microbiome over the rumen microbiome (Supplementary Fig. 1C).

Metagenome-guided metaproteomics was conducted to determine if *Bacteroidaceae* proteins were overexpressed within cattle fed *M. japonica*. A total of 244 metagenomic assembled genomes (MAGs) with over 70% completion and under 10% contamination were generated within this study (Fig. 1C), including 14 MAGs from *Bacteroidaceae* (Supplementary Data 1). A *Bacteroides xylanisolvens* MAG (99.5% completion; 0.7% contamination), referred to as $Bx$MAG$_{BOV}$ (Assembly_18_bin.6), was identified in multiple silage-fed fecal samples and contained predicted CAZymes from families known to be active on carrageenan, including GH16 subfamily 17 members (GH16_17), GH82, GH150, GH167 and sulfatases. These CAZymes were organized into a PUL denoted as $Bx$MAG$_{BOV}$ CarPUL. This MAG (or high identity reads mapped towards CarPULs) was only found in samples from cattle fed seaweed (Supplementary Data 2).

Metaproteomics analysis revealed the largest differences in fecal samples from cattle fed 5% *M. japonica*, with more total proteins and higher overall detection levels as determined by summed label-free quantification (LFQ) intensity (Supplementary Fig. 2A). We found 771 proteins which showed significantly different ($p < 0.05$) protein detection (Fig. 1D); of these, 733 had significantly higher protein detection in the fecal samples collected from cattle fed the seaweed silage diet compared to the control group. Identification of MAGs that were attributed to the significantly higher detected proteins in cattle fed *M. japonica*, revealed that $Bx$MAG$_{BOV}$ accounted for 45% of all proteins ($n = 331$; Fig. 1D). Also, $Bx$MAG$_{BOV}$ was among the most highly abundant populations in our analysis and exhibited a 27.7-fold increase in summed protein detected in fecal samples from cows fed the seaweed silage diet (Supplementary Fig. 2B). Of the total 4,420 unique protein groups recovered by our metaproteomic analysis, 549 proteins were taxonomically assigned to $Bx$MAG$_{BOV}$ (Supplementary Fig. 2C). Our metaproteomic analysis revealed that $Bx$MAG$_{BOV}$ CarPUL was almost exclusively detected in the fecal samples of cows fed *M. japonica* (Fig. 1E; Supplementary Data 3).

### Ruminant-associated *Bacteroides* strains display variation in CarPUL structures

Rumen samples collected from cattle in this study were enriched in minimalized media supplemented with either de-starched *M. japonica* water-soluble extract (*Mj*Ex) or commercially available carrageenans (κ, ι, and λ), allowing for isolation of cattle-derived bacterial species capable of growth on carrageenans. Five *Bacteroides* spp. isolates were successfully enriched on ι-carrageenan and *Mj*Ex, of which two were selected for further long-read genome sequencing, identification, and characterization based on sequencing quality: $Bx$Car5$_{BOV}$ and $Bx$Car17$_{BOV}$, isolated on ι-carrageenan and *Mj*Ex, respectively. $Bx$Car5$_{BOV}$ grew to a higher density than $Bx$Car17$_{BOV}$ on all commercial carrageenans (ι, λ, and κ) and the *Mj*Ex (Fig. 2A, B), despite distorted optical densities from κ-carrageenan viscosity. $Bx$Car17$_{BOV}$ did not have noticeable growth on ι-carrageenan, but did grow on other carrageenan sources (λ, κ, and *Mj*Ex). Growth on κ-carrageenan and *Mj*Ex was observed after a 12 h lag period. The selective uptake of *Mj*Ex by $Bx$Car isolates was directly visualized using fluorescently labeled *Mj*Ex (FLA-*Mj*Ex). Both strains were found to internalize FLA-*Mj*Ex, further supporting that these strains can indeed consume polysaccharides from *M. japonica* (Fig. 2A; Supplementary Fig. 3). FLA-*Mj*Ex import is consistent with a selfish-mode of foraging[29,30], and higher in $Bx$Car5$_{BOV}$ (75% after 24 hr) than in $Bx$Car17$_{BOV}$ (5% after 1 day) (Supplementary Fig. 3). Neither growth nor uptake of carrageenan of FLA-*Mj*Ex was observed for the control *B. xylanisolvens* strain – $Bx$XBA1$_{HOM}$.

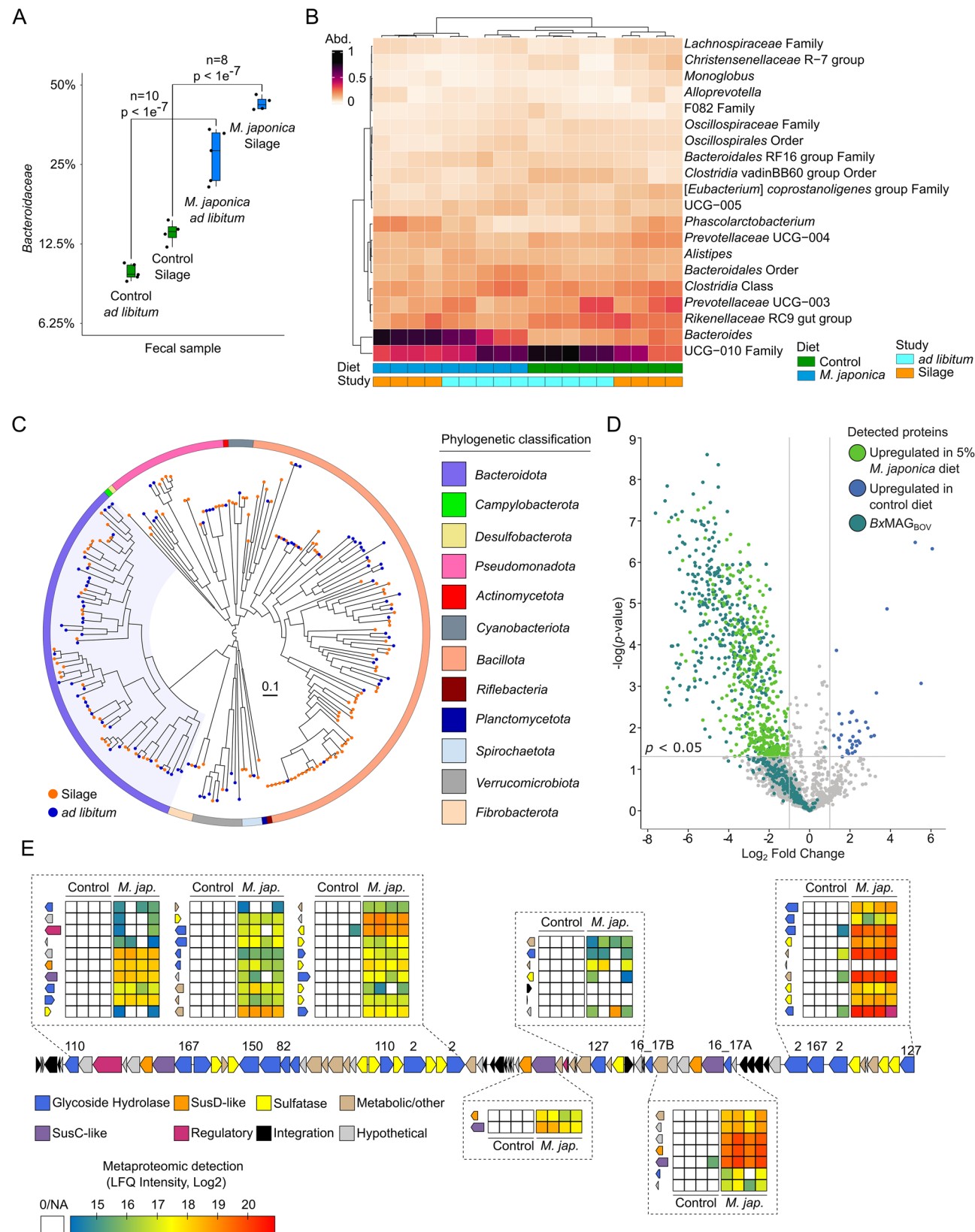

BxCar17_BOV possesses a single short CarPUL, while BxCar5_BOV possesses two CarPULs. One is highly similar to the BxMAG_BOV CarPUL (BxCar5_BOV CarPUL-1), sharing 99-100% identity between sequences (Supplementary Data 4); the second is shorter and unique to BxCar5_BOV (BxCar5_BOV CarPUL-2) (Fig. 2C). Bovine CarPULs contain homologous CAZyme and sulfatase families, differing only in the number of genes present. For example, only BxCar17_BOV contains a single GH16_17; whereas BxMAG and BxCar5_BOV PULs contain two GH16_17s, all sharing between 40-90% identity. Likewise, CarPULs contain one or more copies of sulfatase subfamilies 1_8, 1_16, 1_30, and 1_81 (>80% shared identity; Supplementary Data 4). The exception is BxCar5_BOV CarPUL-2, which is missing copies of S1_15 and S1_20.

**Fig. 1 | *M. japonica* supplementation increases *Bacteroides* within the fecal microbiota. A** Percentage abundance change of *Bacteroidaceae* between diet and study fecal samples. A linear regression model of % composition was generated using microviz[64], and a Tukey test used to measure significance (adjusted *p* value; two-sided)[79]. **B** 16S rRNA gene relative abundance of genus' above 10,000 reads within fecal samples. Diet (Control: *n* = 9 and *M. japonica*: *n* = 9) and Study (*ad libitum*: *n* = 10 and Silage: *n* = 8) are denoted below the heatmap. **C** MAGs generated and dereplicated between fecal and rumen metagenomes. Rumen MAGs are denoted with an orange circle and fecal with blue. **D** Volcano plot showing proteins with significantly different protein expression recovered from fecal samples from cattle fed diet with 5% inclusion level of *M. japonica* vs those fed the control diet (two-sided Student's T-test; S0 = 0). Difference in protein expression is displayed as Log$_2$ fold change in LFQ intensities. Proteins are colored based on their presence within *Bx*MAG$_{BOV}$ genome and the *Bx*MAG$_{BOV}$ CarPUL. Critical and highly expressed proteins are further labelled. **E** *Bx*MAG$_{BOV}$ CarPUL detected within *M. japonica* supplemented cattle. Heatmaps indicate LFQ intensity for each *Bx*MAG$_{BOV}$ CarPUL gene for all four replicates of fecal samples of cattle fed control diet and diet supplemented with 5% *M. japonica* in the silage study. Genes that do not have a heatmap present were not identified within proteomes. Source data are provided as a Source Data file.

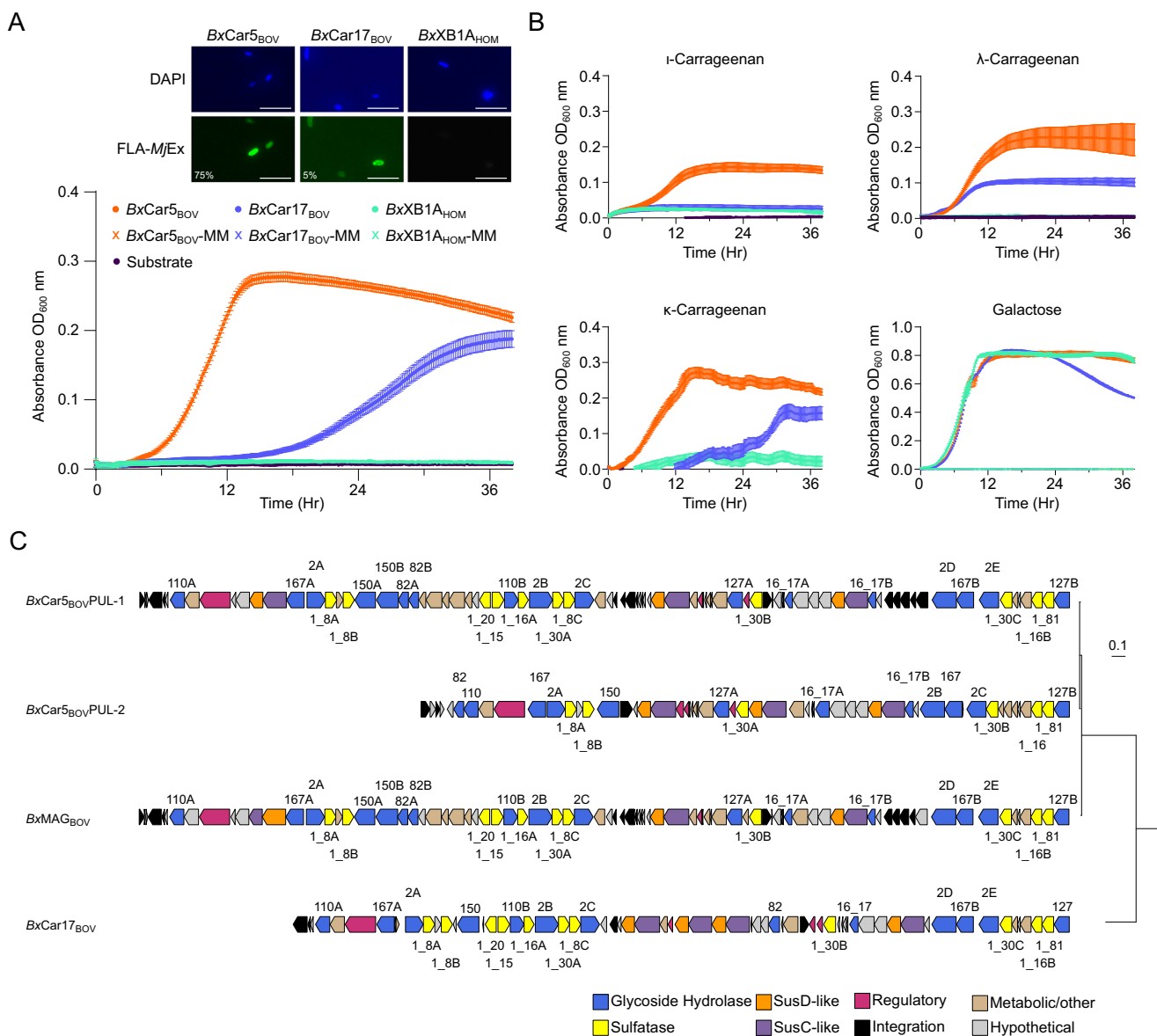

**Fig. 2 | *B. xylanisolvens* enriched from *M. japonica* supplemented cattle contain variations on the *Bx*Car$_{BOV}$ PUL. A** Top: Isolates were incubated with 0.2% fluorescently labeled *M. japonica* extract (FLA-*Mj*Ex) for 1 d. Samples were co-stained with DAPI and imaged using an epifluorescence microscope (LED light cubes DAPI (EX: 385/30 EM: 450/50 DM: 425, ED light cubes FLA-MjEx (EX: 470/40 EM: 525/50 DM: 495). Isolates were primed for 24 hr on *Mj*Ex. Scale bar = 5 μm. Values in bottom left corner represent the percentage of cells showing FLA-*Mj*Ex uptake. Bottom: *B*. *xylanisolvens* isolates and control *B. xylanisolvens* XB1A grown on 0.3% *Mj*Ex (*n* = 3; error bars represent minimum and maximum OD$_{600}$ nm). **B** *Bacteroides* isolates grown on 0.3% commercial carrageenans. OD$_{600}$ nm was observed every 10 min (*n* = 3). Negative OD values were excluded from the plots. **C** Conservation of *Bx*Car5$_{BOV}$ CarPUL−1 & CarPUL-2, *Bx*Car17$_{BOV}$ and *Bx*MAG$_{BOV}$ CarPULs aligned via OrthoFinder species tree.

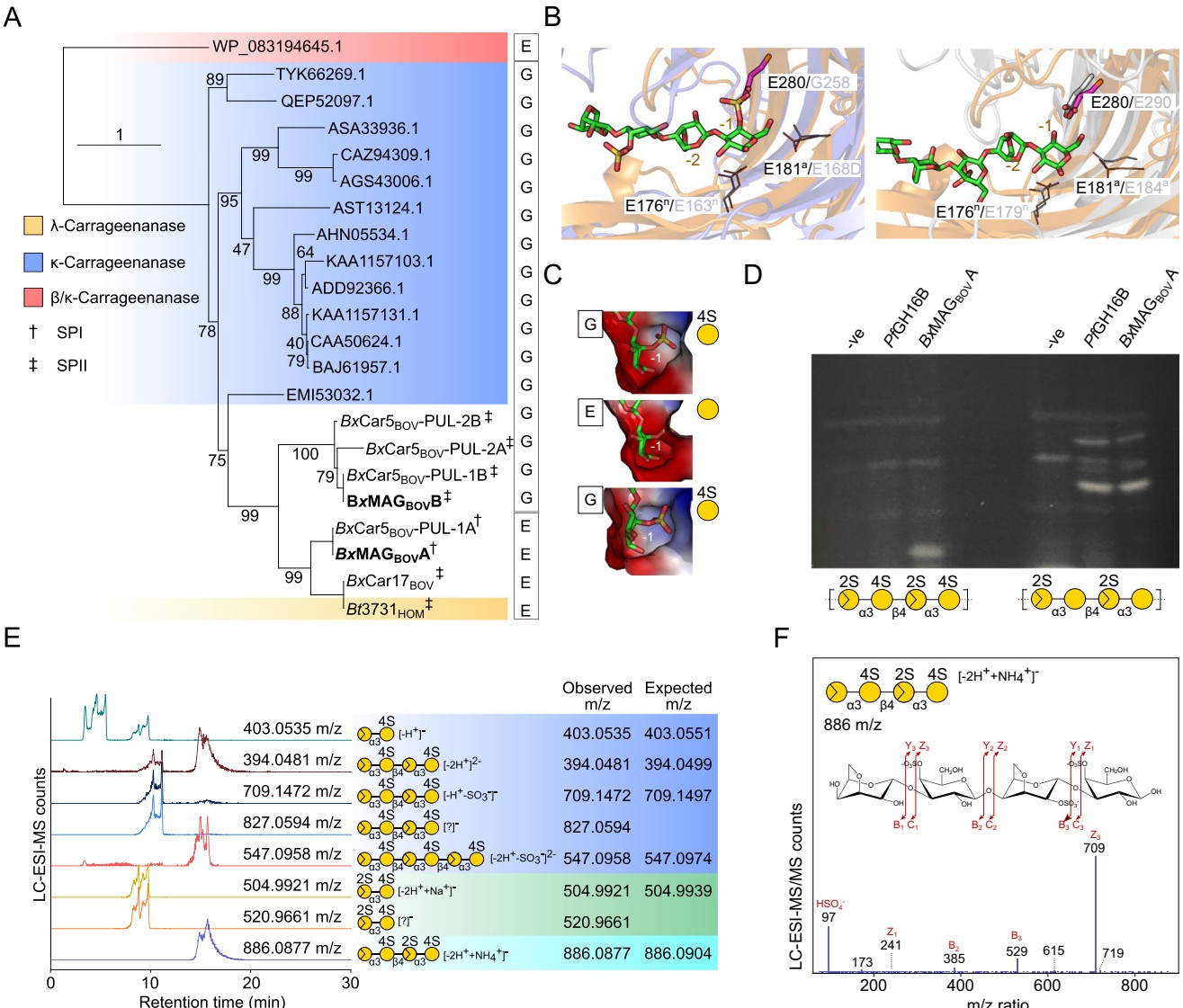

**Fig. 3 | *Bx*MAG GH16_17 form functionally distinct clades to saccharify carrageenan subtypes. A** SACCHARIS v2 generated phylogeny of GH16_17 members from *Bx*MAG_BOV, and *Bx*Car5_BOV and *Bx*Car17_BOV within this study aligned to characterized GH16_17 members from the CAZy database (www.cazy.org). Signal peptides of *Bx*Car enzymes are denoted by † (SPI) and ‡.(SPII) **B** Structural analysis of *Bx*MAG_Bov GH16_17A. Left: Overlay of the structure of *Bx*MAG_BOV GH16_17A (orange) with the GH16_17 κ-carrageenase from *Pseudoalteromonas carrageenovora* (blue; PDB ID 5OCQ). Key residues within only the -1 and -2 subsites are shown as sticks with the residues acting as nucleophile or acid/base labeled and indicated with superscript "n" or "a", respectively. The substitution in *Bx*MAG_BOV GH16_17A blocking accommodation of the 4S group is shown as a magenta stick. Right: Overlay of the structure of *Bx*MAG_BOV GH16_17A (orange) with the GH16_13 β/κ-carrageenase from *Wenyingzhuangia fucanilytica* (gray; PDB ID 9INT). **C** Surface representations of the -1 subsites of 5OCQ (top), *Bx*MAG_BOV GH16_17A (middle), and

an AlphaFold 3[34] generated model of *Bx*MAG_BOV GH16_17B, respectively. The ligand in 5OCQ is shown; ligands in *Bx*MAG_BOV GH16_17A and the *Bx*MAG_BOV GH16_17B model are derived from overlays with 5OCQ. **D** The activity of *Pf*GH16B (an α/β-carrageenan specific GH16_13)[35] and *Bx*MAG_BOV GH16_17A on ι-carrageenan (Left) and on ι-carrageenan pretreated with *Pf*S1_19B to remove 4S modifications (Right) as examined by fluorophore-assisted carbohydrate electrophoresis (n = 3). Gels were processed in parallel. **E** *Bx*MAG_BOV GH16_17B digests of *Mj*Ex were analyzed by LC-ESI-MS, and example extracted ion chromatograms are shown. **F** ESI-MS/MS with HCD was performed in order to identify the extracted ions. The MS2 product ion spectra shown are consistent with neo-κ/ι-hybrid tetrasaccharide, the majority enzymatic product from the *Mj*Ex. Monosaccharide symbols are displayed according to the Symbol Nomenclature for Glycans system[101]. Source data are provided as a Source Data file.

## Phylogenetically distinct GH16_17 members have differing carrageenan specificities

Keystone CAZymes play critical roles at the surface of microorganisms, and differences in their specificity or catalytic efficiency may dictate saccharification cascades of carrageenan subtypes and growth propensities. To elucidate functional differences in keystone GH16_17 members, primary sequences were aligned and visualized using SACCHARIS v2[31]. Three distinct clades were observed (Fig. 3A). *Bx*Car17_BOV GH16_7A shares homology to a previously characterized λ-carrageenase from *Bacteroides thetaiotaomicron* 3731 (*Bt*3731)[20];

whereas *Bx*MAG_BOV and *Bx*Car5_BOV GH16_17 enzymes partitioned with members from *Bacteroides ovatus* CL02T12C04 (*Bo*12C04_HOM). Notably, catalytic residues were conserved in all sequences (Supplementary Fig. 4A, B). To determine if clades were associated with unique carrageenan subtype specificities, *Bx*MAG_BOV GH16_17A and GH16_17B were selected for biochemical characterization.

The X-ray crystal structure of *Bx*MAG_BOV GH16_17A was solved to 2.0 Å resolution (PDB ID: 9EFL). The protein crystallized with a single monomer in the asymmetric unit. An analysis of interactions in the crystal lattice using PISA indicates a lack of stable quaternary

structures. Superimposition with the GH16_17 *Pseudoalteromonas carrageenovora* CgkA, in complex with a κ-carrageenan oligosaccharide[32], revealed conservation of catalytic residues in the -1 and -2 subsites. A discriminating feature was the substitution of G258 in CgkA with E280 of $Bx$MAG$_{BOV}$ GH16_17A, resulting in closure of a pocket responsible for 4S-Gal accommodation and introduces a steric clash with the active site surface (Fig. 3B). This structural determinant is functionally conserved in the GH16_13 κ/β-carrageenase from *Wenyingzhuangia fucanilytica*[33], and unequivocally selects for unsulfated galactose in the -1 subsite (Fig. 3B). To compare clade-dependent specificity, an AlphaFold 3[34] generated model of $Bx$MAG$_{BOV}$ GH16_17B displayed the same active site architecture as CgkA complete with the 4S-binding pocket (Fig. 3C; and Supplementary Fig. 5). Therefore, we hypothesized that $Bx$MAG$_{BOV}$ GH16_17A may have activity on partially desulfated/hybrid carrageenans, similar to the GH16_13 enzymes *W. fucanilytica* and *Pseudoalteromonas fuliginea* PS47 ($Pf$GH16B). Specifically, the activity of $Pf$GH16B required prior removal of 4S from κ/ι-carrageenan for hydrolysis of the backbone[35]. Using FACE to detect the products of ι-carrageenan hydrolysis by $Bx$MAG$_{BOV}$ GH16_17A, we found that native ι-carrageenan was not hydrolyzed by $Pf$GH16B and was a poor substrate for $Bx$MAG$_{BOV}$ GH16_17A (Fig. 3D). However, pretreatment of the carrageenan with a endo-4S-sulfatase, $Pf$S1_19B[15], rendered it an improved substrate for $Bx$MAG$_{BOV}$ GH16_17A, producing the same product profile as $Pf$GH16B. These results support the assignment of $Bx$MAG$_{BOV}$ GH16_17A as a carrageenase active on hybrid substrates with full or partial removal of the 4-sulfate groups.

In contrast, $Bx$MAG$_{BOV}$ GH16_17B demonstrated endo-activity on $Mj$Ex and all commercially available forms of carrageenan, including κ- and ι-carrageenan and κ/ι-carrageenan hybrids (Fig. 3E, F; and Supplementary Fig. 6A–D). Although digestion of λ-carrageenan was observed via TLC (Supplementary Fig. 6E), these products could not be confirmed by LC-MS and any products resulting from this commercial substrate were exclusively κ- and ι-carrageenan oligosaccharides. This supports that $Bx$MAG$_{BOV}$ GH16_17B is an endo-κ/ι-carrageenase. Treating $Mj$Ex with $Bx$MAG$_{BOV}$ GH16_17B also improved the growth of the $Bx$Car17$_{BOV}$ isolate, which lacks a homologous enzyme (Supplementary Fig. 6F), suggesting it has a biological role as a keystone enzyme in κ/ι-carrageenan utilization.

### CarPULS in other herbivorous artiodactyls
CarPULs have previously been identified within human and captive great ape GIT microbiomes[36], which showed as high as 100% sequence identity towards the bovine sourced CarPUL sequences (Fig. 4; and Supplementary Data 5). CAZyme BLAST hits for GH16_17 members performed here highlight that carrageenan utilization is widespread amongst geographically distinct human GIT datasets (Fig. 4A); however, no ruminant homologs were identified during our database analysis. Therefore, to investigate if CarPULs were present in other ruminant microbiomes, or if proliferation into bovines was a rare event, we analysed available artiodactyl metagenomic datasets. ~17 Tb of shotgun metagenomic read datasets from buffalo, cattle, deer, moose, sheep, and yak collected from public databases (Supplementary Data 6) were searched against CarPUL protein sequences using BLAST to identify high similarity targets. Sequence similarity varied throughout the CarPULs and hosts; however, many regions demonstrated high conservation ( > 90%) in the keystone GH16_17 enzymes. In particular, reads from buffalo, cattle, and deer metagenomes contained frequent hits with 90–100% sequence identity towards GH16_17 members within $Bx$MAG CarPUL and $Bx$Car5 CarPUL-1. Buffalo metagenomic read sets contained the highest identity hits towards all CarPULs, sharing homology with other carrageenan active families GH82 and GH150 (Supplementary Data 6). Metagenomic datasets from Chinese Water Buffalo microbiomes[37] displayed sequences with 100% identity to GH16_17 members and were selected for co-assembly using MegaHIT to assemble full-length GH16_17 members. Open reading

frames originating from this assembly yielded multiple sequences (Meta$_{BUB}$) that were closely related with CarPUL GH16_17 enzymes identified in this study.

In addition, fecal samples from herbivorous artiodactyls housed at the Wilder Institute/Calgary Zoo (Supplementary Data 7) were used for culture-enriched metagenomics and selective isolations with $Mj$Ex. Culture-enriched metagenomics identified carrageenan-active CAZyme members within musk deer and giraffe, which was further supported through isolation of a *B. zhangwenhongii* strain from giraffe ($Bz$Car$_{GIR}$) and *B. xylanisolvens* isolated from musk deer ($Bx$Car$_{MOS}$). The CarPULs and CAZymes from these sources showed high synteny and sequence relatedness with CarPULs found in cattle from this study and human-associated strains (Fig. 4B). Interestingly, the giraffe-derived *B. zhangwenhongii* CarPUL demonstrated near 100% identity with the $Bt$3731 CarPUL (Supplementary Data 5).

### Shared homology towards marine, sediment, and fish gut microorganisms
The wide distribution of CarPULs in terrestrial vertebrate GIT microbiomes raises an intriguing question about the origins of these CAZymes: did they evolve de novo in terrestrial vertebrate guts under dietary selection or originate from transfer from marine microorganisms, a phenomenon described as the "sushi factor" for the porphyran metabolism in the GIT microbiomes of Japanese people[18]. To explore this question, we compared sequence similarity between the mammalian CarPUL genes and NCBI-sourced marine and environmental genes and genomes and the *M. japonica* surface-associated microbiota ($Mj$SM) collected in this study through scraping of *M. japonica* tufts. The top 100 BLAST hits and previously characterized CAZyme members[38] were aligned to CarPUL sequences. Homology was observed between CarPUL genes, and marine pelagic and sediment microorganisms within the GH2 family (Supplementary Fig. 7). The highest identity between ruminant-sourced CarPULs and the $Mj$SM was observed at the gene level between sulfatases (between 70-84% identity for sulfatase 1_30 *vs.* $Mj$SM; Supplementary Data 8), with identity above 50% existing between GH2, GH110, and GH167 members. These homologs were not detected in the $Mj$SM metaproteome suggesting these genes were not expressed (Supplementary Data 3). The highest identity towards CarPUL sequences was found in *Rikenellaceae* MAGs assembled from *Kyphosus sydneyanus* (Silver drummer fish) hindgut microbial communities[39]. SACCHARIS phylogenies revealed *Rikenellaceae* MAG GH16_17 members partitioned with CarPUL GH16_17 members, forming two distinct clades between $Bx$Car5$_{BOV}$ PUL-1 GH16_17A & B (Fig. 5A). Nearly the entire $Bx$Car5$_{BOV}$ CarPUL-2 shared homology with *R. alistipes* MAGs (Fig. 5B), with the highest identity between putative sugar kinases (92%; Supplementary Data 9). Genes surrounding the kinase (including metabolic genes, sulfatases, and GH127 members) were also highly syntenic with *R. alistipes* contigs (Fig. 5C).

## Discussion
### Spatial digestion of *M. japonica* in bovine GIT
We saw a significant increase in *Bacteroidaceae* abundance within the fecal microbiome of cattle, underpinning carrageenan digestion may be a more pronounced trait within the lower GIT. *Bacteroidota*, primarily *Prevotella*, are widely recognized as one of the most abundant fiber degrading bacterial phyla in the rumen[26,40]. Often overlooked, however, is the prowess of *Bacteroides* within the hindgut of ruminants that forage on undigested feed residues that bypass the rumen[41], such as cellulose and hemicelluloses[42,43]. This can be attributed to increased bulk within the digestive tract[39], which could be impacted by seaweed polysaccharides as observed in the digesta of mice[44]. Metagenome-guided metaproteomics analysis presented here suggests that *Bacteroides*-affiliated MAGs are one of the primary mechanisms for microbial degradation and utilization of *M. japonica* carrageenans in

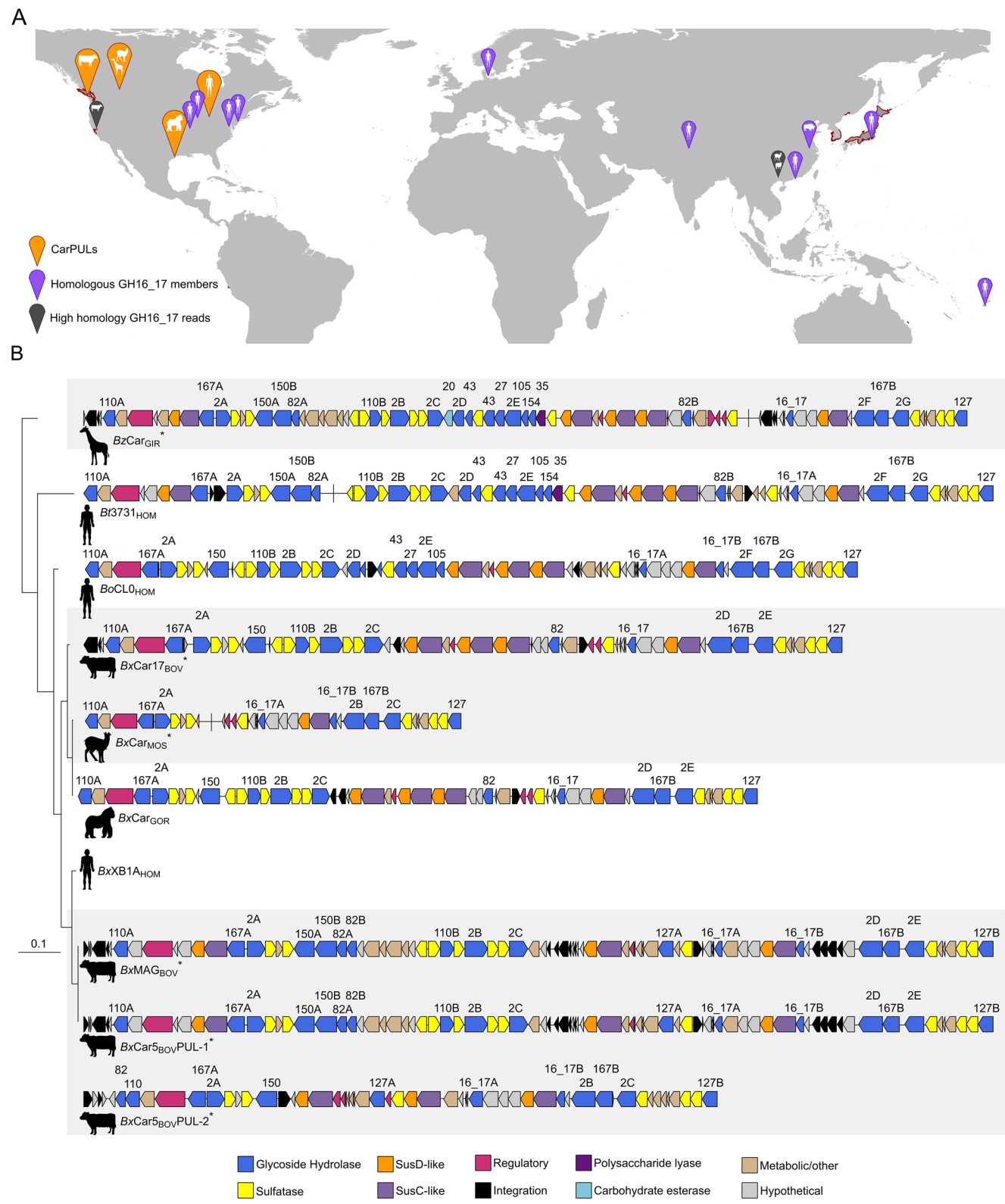

**Fig. 4 | *Bacteroides* carrageenan PULs and CAZymes display homology between host species and geographic regions. A** CarPULs and orphaned CarPUL genes identified within mammalian GIT metagenomes. Identified CarPULs, GH16_17 homologs, and highly homologous reads (100%) of putative GH16_17 members are marked on the map. Red outlines on the map indicate habitats of *M. japonica*[100]. The map was generated using ggOceanMaps[102]. **B** CarPUL diagrams of bovine *B. xylanisolvens* (this study), giraffe (*Bz*Car_GIR) and musk deer (*Bx*Car_MOS) *Bacteroides* (this study) and carrageenan PULs found within human (*Bt*3731_HOM; *Bo*CL0_HOM; *Bx*Xba1_HOM)[20] and gorilla (*Bx*Car_GOR)[36] associated *Bacteroides*.

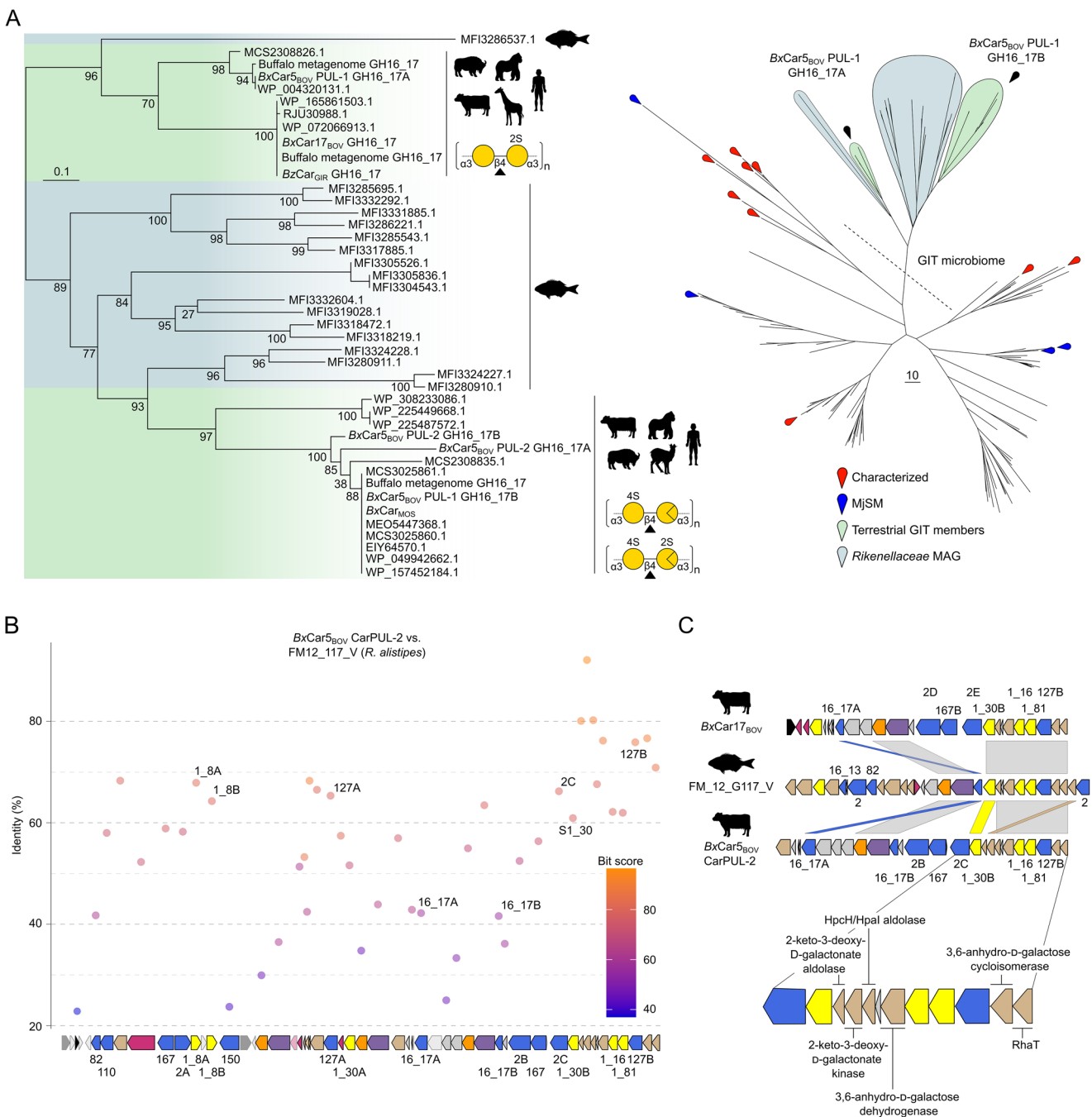

**Fig. 5 | *Bx*Car PULs show homology with CarPULs from the marine fish GIT-associated *Rikenellaceae* spp. A** The top 100 BLAST hits towards each CarPUL GH16_17 member, predicted *Mj*SM GH16_17 members, and characterized GH16_17 members with the database were pooled alongside CarPUL GH16_17s and a phylogeny was created of aligned predicted catalytic domains. Clades are colored on their host habitat; Green: terrestrial vertebrate *Bacteroides* spp. and teal: *K. sydneyanus* GIT members (*Rikenellaceae* MAGs; Facimoto et al. 2024). Individual nodes are denoted by a pin. BLAST hits are detailed in Supplementary Data 9. A dotted line indicates clades exclusively containing GIT microbiome members (terrestrial vertebrate or *K. sydneyanus* sourced) **B** *Bx*Car5$_{BOV}$ CarPUL-2 genes were BLASTed against *Rikenellaceae* MAGs. Shown are the top hits towards FM_12_G117_V (*R. alistipes*). **C** CarPUL synteny towards *Rikenellaceae* MAGs (shown FM_12_G117_V). Regions of high synteny and homology were further expanded to demonstrate core functionality of CarPUL ancestors. Source data are provided as a Source Data file.

the lower GIT of cattle (Fig. 1; and Supplementary Fig. 2C). There was a minor increase in ruminal *Bacteroidaceae* under *M. japonica* supplementation, which paled in comparison to the increase in fecal samples. This is likely due to *Bacteroides* spp. being more abundant within the lower GIT[26] (Supplementary Fig. 1A). These findings suggest the disparity between research on the rumen microbiome compared to the lower GIT microbiome, may hamper the discovery of seaweed degrading pathways within ruminants. Further, the impact carrageenans, and potentially other seaweed polysaccharides, has on the lower

GIT microbiome may provide targeted prebiotic benefits or function as delivery systems to the hindgut of ruminants. It should be noted that *M. japonica* supplementation was shown to be digestible in ruminants, meeting nutritional crude protein requirements; however, not it did not significantly reducing methane emmisions[45].

## CarPULs are structurally and functionally diverse latent traits
Although CarPULs were detected within many mammalian GIT microbiome datasets, they were not detected in control animals from

the *ad libitum* or silage studies. This suggests that either CarPULS are not universal within herbivorous animals or sequencing depth may be insufficient to detect low-abundance PULs and CAZymes (*i.e.*, detection requires enrichment). Carrageenan depolymerization requires a large complement of CAZymes, sulfatases, and metabolic genes which is reflected in the structural heterogeneity of CarPUL structures (Fig. 2C, 4B). Although there is potential for other enzymes to be involved in carrageenan utilization, the CarPULs discovered here appear to contain all the required catalytic machinery[46]. No other CAZymes belonging to known carrageenan-specific families were identified in the genomes and metagenomes, or upregulated within the metaproteomic datasets (Supplementary Data 3). Together these results suggest that CarPULs encode the conserved core machinery for carrageenan utilization within *Bacteroides* species.

Ruminants are notorious for the functional capacity of their rumen microbial communities. The role of latent traits in dietary adaptation extends this paradigm beyond what is detectable by low-depth, high-coverage sequencing methods of non-adapted metagenomes. It is also possible that bespoke dietary enrichment strategies are required to detect cryptic latent traits. Red seaweeds are primarily composed of sulfated galactans in the form of agars and carrageenans, which can be modified with *O*-methylation, and discriminatory sulfation patterns[47]. Carrageenans are not chemically pure in the cell walls of red seaweeds and can exist as hybrids or mixtures, which influences their physiological properties[48]. Linkage analysis conducted here indicates that *M. japonica* matrix polysaccharides primarily consists of 'kappa family' (κ-, ι-, ν-, μ-) carrageenans[47,49] (Supplementary Fig. 8; Supplementary Method 1; Supplementary Note 1). Further, structural diversity in dietary seaweeds can be attributed to variations in algal species or seasonal factors[47].

*Rikenellaceae* CarPULs from the *K. sydneyanus* GIT likely saccharify κ/ι-carrageenans found within *Caulacanthus ustulatis* and other dietary seaweeds[50,51]. Similarly, *M. japonica* may not have been a dietary component of ancestral ruminants and a primary source of carrageenans for the animals that were sampled. *M. japonica* is an introduced seaweed to the Pacific Northwest and originates from coastal waters of the Northwest Pacific (Korea, Japan, Russia)[52]; it is not present in all locations where GH16_17 homologs were identified (Fig. 4A). Mammalian GIT CarPULs can display different architectures and harbor putative CAZymes which target multiple carrageenan subtypes[35,46], suggesting these pathways may be suited to target a diversity of carrageenans in nature, with strain diversity contributing to individual catabolic responses[53].

## GH16_17 phylogeny dictates specificity for carrageenan subtypes

CarPUL GH16_17 enzymes were observed to form phylogenetically distinct clades that correlated with substrate specificity and not host origin, with two distinct clades containing sequences from ruminant and non-ruminant hosts (Fig. 4, 5). *Bx*Car5$_{BOV}$ and *Bo*12C04$_{HOM}$ each contain two paralogs of GH16_17, one specific for κ/ι-carrageenan and one for α-carrageenan (Fig. 3), a product of ι-carrageenan 4S-Gal*p* desulfation[35]. This was validated by their growth in pure culture on κ/ι-carrageenan[20] (Fig. 2B). Furthermore, the *Bx*Car17$_{BOV}$ GH16_17 is 100% identical to that found in the CarPUL from *Bt*3731$_{HOM}$ (Supplementary Data 5), a strain which was isolated from humans and shown to metabolize λ-carrageenan[20]. In support, *Bx*Car17$_{BOV}$ grew on λ-carrageenan, and displayed compromised growth on *Mj*Ex and κ-carrageenan, requiring a 12 h lag period (Fig. 2A, B).

A molecular determinant between κ/ι-carrageenan- and α-carrageenan-active GH16_17 enzymes result from mutations within the active-site pocket (Fig. 3C; and Supplementary Fig. 4A & B). *Rikenellaceae* GH16_17 members did not display the glycine to glutamate mutation that excludes 4S-Gal*p* within the active site. In many cases, GH16_17 genes are paired with a GH16_13, which could provide a

surrogate α-carrageenanse activity (Fig. 5C). Intriguingly, the one *Rikenellaceae* MAG (FM_16_G121_IV) that did not contain a predicted GH16_13 member, possessed a GH16_17 with the highest similarity to *Bx*MAG$_{BOV}$ GH16_17A; however, in this case the glycine to glutamic acid mutation was not observed. This may provide a transitional snapshot into the evolutionary landscape of these partnered activities. The presence of multiple *Bx*MAG$_{BOV}$ GH16_17 members within this clade suggests that gene duplication and neofunctionalization of the GH16_17 subfamily may have occurred, similar to what was described for polysaccharide lyase family 2[54]. The presence of GH16_17 members with complimentary activities suggests it is a foraging strategy for complete saccharification of carrageenan-rich seaweeds.

## CarPUL evolution and radiation into diverse microbial ecosystems

Previously, ancient HGT events have been implicated in the evolution of alginate utilization loci in human[24] and bovine[47] GIT metagenomes. Here, wide-spread radiation of CarPULs was observed between mammalian-associated metagenomes and isolates, and NCBI sequences across geographic barriers and host microbiomes. *Bacteroides* spp. compared herein contain highly syntenic CarPULs, irrespective of host source (giraffe, musk deer, human, cattle) and geographic region (Alberta, CA; Vancouver, CA; Michigan, US) (Fig. 4B). Furthermore, high identity was observed between CarPUL CAZymes and GIT metagenomic reads from buffalo, sheep, and water buffalo from farms in Chinese provinces[37,55], and cattle sampled on American farms[56] (Fig. 4A; and Supplementary Data 4). The geographic and host distribution of CarPULs suggests that these loci were acquired by ancestral seaweed-foraging terrestrial vertebrates and then maintained with low sequence divergence. Hierarchical responses to dietary seaweed polysaccharides and functional specialization of CarPULs at the strain or community level may help inform the origin and maintenance of latent traits. In this regard, we also discovered a porphyran/agarose PUL syntenic to those found within human gut *Bacteroides* spp. in the enriched giraffe fecal metagenome (Supplementary Fig. 9)[20]. As these catabolic abilities may be hidden in the genetic dark matter[25], assumptions of their presence or absence in the microbiome should be made cautiously. Of interest, CarPULs may have penetrated other ecosystems, as GH2 members discovered in Canadian landfills cluster with *K. sydneyanus* GIT and terrestrial vertebrate GIT members (Supplementary Fig. 7).

Bovine *Bacteroides* spp. CarPUL genes display high sequence identity with CarPULs from *R. alistipes* MAGs within the GIT of *K. sydneyanus*. This pattern was observed for all bovine CarPUL CAZyme families; and most notably for GH2, a well-studied, exo-acting polyspecific family. In the GH2 phylogenetic tree, nearly all CarPUL-associated sequences were neighbored by *R. alistipes* enzymes, with many also flanked by GH2 members from marine pelagic and sediment microorganisms (Supplementary Fig. 7). These findings agree with previous reports that CarPULs in GIT-associated microbiomes originated by HGT from marine microoganisms[20]. Although homologs of ruminant-sourced CarPUL genes were found within marine-source sequence datasets and the *Mj*SM, there was not convincing evidence to conclude HGT from these microorganisms (Supplementary Data 8). If a closer relative of bovine CarPULs exists within an as-yet-undiscovered marine bacterium, it may remain difficult to find. Marine environmental microbiomes contain great microbial diversity skewed by seasonal variation and other environmental stressers[57]. Therefore, the "ancestral" CarPUL may only be abundant during seasonally- or geographically-constrained windows in extant bacteria, and absent from the available sequence datasets.

The origins of GIT-associated CarPULs appear complex. The mean % GC content within the bovine CarPULs was significantly different to the remainder of their genomes, suggesting the presence of these pathways resulted from HGT (Supplementary Fig. 10; Supplementary

Method 2; Supplementary Note 2). In contrast, *R. alistipes* MAG CarPUL genes were very similar in mean % GC content to the remainder of their genomes, suggesting CarPUL acquisition in the GIT of fish occurred much earlier than in terrestrial ruminants. This would correspond with *K. sydneyanus* feeding on seaweed well before the emergence of seaweed-consuming land animals. The differences in GC content also suggest acquisition occurred as separate events. However, it does not rule out that fish GIT microorganisms may share a common origin with terrestrial mammalian CarPULs. CarPUL genes could have co-evolved throughout the vertebrate lineage or transferred into mammalian GIT communities during predation or fecal-oral transfer. In the latter case, HGT exchange between *Bacteroidales* Order members has been observed within the human GIT[58] and marine ecosystems[59]. It is therefore likely that fish GIT *Bacteroidales* members, such as *Rikenellaceae*, may contain functional HGT genetic elements compatible with those found in terrestrial mammalian GIT members. Overall homology between putative integrative elements within bovine GIT CarPULs and fish GIT CarPULs was low; however, select genes do share up to 72% identity (IS4 family transposase; Supplementary Data 9). Deeper investigation into the GIT microbiota of fish or other marine animals that forage on red seaweeds may hold subsequent clues into the mechanisms and origins of terrestrial mammalian GIT seaweed metabolism. Regardless of their origins, it is clear that CarPUL function is not host or geographically restricted and has proliferated within terrestrial mammalian GIT microbiomes as a latent trait within the genetic dark matter.

## Methods

### Ethics approval
All procedures and protocols involving cattle were reviewed and approved by the Thompson Rivers University animal care committee (award #: 101948) for following the guidelines of the Canadian Council on Animal Care. Fecal collection from animals at the Wilder Institute/Calgary Zoo was approved by the Wilder Institute/Calgary Zoo Animal Welfare, Ethics, and Research Review Committee (Protocol 2023-06).

### Animals and environment sampling
Two animal trials were conducted as part of this study, *ad libitum* feed trial and a controlled silage trial. Both trials were conducted at a certified organic farm (Beaver Meadows farm - #03-16-198). For the *ad libitum* trial, Cattle (n = 10; mixed heifers and steers) were separated into two groups on pasture. In one pasture, *M. japonica* was offered *ad libitum* (ca. 1 lb/cow). Fresh fecal samples were collected and rumen samples were gathered from 10 animals (5 control, 5 treatment) slaughtered at Gunter Bros Meat Co. Ltd, 6200 Ledington Road, Courtney, BC. For the silage trial, Angus X Hereford cattle (n = 4; mixed heifers and steers) were fed a backgrounding diet (bait feed: alfalfa pellets) for two weeks, before switching to a grass-based silage. Control cattle were fed a grass silage diet for two weeks (Supplementary Table 1). After the two-week period, fresh fecal samples for each animal were collected and rumen samples were collected via stomach tubing. The diet was then changed to include a 5% *M. japonica* supplement for another two-week period (Supplementary Table 1). Fresh fecal samples were collected and rumen samples were collected via oral stomach tubing.

*M. japonica* samples were collected from the shoreline on Maple Guard Dr, British Columbia, Canada (Lat 49.441695, Long -124.676979). *M. japonica* tufts were rinsed in artificial seawater to remove transient microbes, then finely scrapped with a sterile scalpel into 15 mL of artificial seawater and was passed through a 0.2 μm filter. All samples were directly frozen in LN₂ and stored.

### Metagenomic sequencing and data analysis
DNA from all samples were extracted using the DNeasy PowerSoil Pro kit (Qiagen, Canada) according to the manufacture's protocol. The DNA quality was analysed using Nanodrop 2000. 16S rRNA amplicon sequencing was achieved using Illumina MiSeq 2500 (Génome Québec). The 16S rRNA gene sequences were split into individual samples through the internal barcode. Sequences were trimmed using Trimmomatic (v0.39)[60] and passed through Kraken2 (v2.1.2 SILVA NR99 database)[61] for annotation. Kraken-biome[62] was used to create a biome file for phyloseq (v1.48.0)[63]. Statistical analysis was done using the micoViz R package (v0.12.4)[64], dplyr (v1.1.4)[65], and ggplot2 (v3.5.1)[66].

DNA from fecal and rumen samples from the *ad libitum* study and the silage study, as well as the *Mj*SM were sequenced using Illumina Novaseq 6000 150 bp PE reads (Génome Québec). Raw reads were quality trimmed using Trimmomatic (SLIDINGWINDOW:5:20) and assembled individually using MetaSPAdes (v3.13.0)[67], and biological replicates together in a co-assembly using MEGAHIT (v1.1.3)[68] (Supplementary Data 7). Contigs were binned and refined using the MetaWRAP (v1.3.2) binning/refinement modules[69], with metaBAT2 (v2.12.1)[70], MaxBin2 (v2.2.6)[71], and CONCOCT (v1.1.0)[72]. Bins/MAGs between individual assemblies and co-assemblies were merged and de-replicated with dRep (v3.0.0)[73] at 70% completion and <10% contamination.

### Metaproteomic data generation
To collect microbial cells from sample material with fibrous debris, fecal samples as well as the environmental samples (soil, compost, and filter paper of *M. japonica*) were gently washed prior to lysis and protein extraction. Microbial cells in 0.5 g of sample (fecal or environmental) were dissociated from the sample material through cycles of centrifugation and addition of dissociation buffer containing tert-butanol, Tween 80 and HCl to a pH of 2, and the cell containing supernatant collected and combined after each cycle[74]. The supernatant was centrifuged to a cell-containing pellet, which was further washed in a cell wash buffer containing 10 mM Tris-HCl (pH=8) and 1 M NaCl. The remaining cell pellet was resuspended in lysis buffer (30 mM DTT, 150 mM Tris-HCl (pH=8), 0.3% Triton X-100, 12% SDS).

Similarly, 300 μl of the abovementioned lysis buffer was added to 0.5 g of rumen fluid. Glass beads (4 mm: ≤ 160 μm) were added to all samples (fecal, environmental and rumen samples), briefly vortexed and let rest on ice for 30 minutes. Cells were mechanically lysed using FastPrep-24 Classic Grinder for 3 cycles of 60 s at 4.0 meter/second. Samples were further centrifuged at 16,000 × *g* for 15 min at 4 °C and lysate transferred to a clean tube. For clean-up purposes, lysates were run on SDS-PAGE using Any-kD Mini-PROTEAN TGX Stain-Free gels and stained with Coomassie Blue R-250. Visible bands on gels were carefully excised and reduced, alkylated, and digested to peptides with trypsin. Peptides were analysed by nano-LC/MSMS using a timsTOF Pro mass spectrometer (Trapped Ion Mobility Spectrometry quadrupole time-of-flight mass spectrometer) (Bruker, Germany).

### Metaproteomic data analysis
MS raw data were analysed using FragPipe, powered by the proteomic search engine MSFragger[75] along with Philosopher[76] and IonQuant[77] for post-processing of MSFragger results including label-free quantification (LFQ) and false discovery rate (FDR) filtering. A sample specific sequence database used consisted of MAGs recovered from the silage study as well as MAGs from *Mj*SM. In total 443 MAGs (> 70% completion and <10% contamination). The MAG database was supplemented with common contaminant protein entries, such as human keratin, and decoy protein entries based on reverse sequences for estimation of false discovery rate (FDR). For LFQ, using IonQuant, the Match-between-runs (MBR) feature was enabled. Oxidation of methionine and protein N-terminal acetylation were used as variable modifications, while carbomidomethylation of cysteine residues was used as a fixed modification. Trypsin was chosen as digestive enzyme and max missed cleavages allowed was set to one. FDR and MBR-FDR were both set to 1%. Further, detected protein groups were explored in Perseus (v.

2.0.7.0)[78]. Protein groups identified as possible contaminants were removed. Proteins were considered valid if they were detected in least 2/4 replicates, or 2/5 replicates for the environmental samples, in at least one sample type group (fecal samples from cows fed the control silage diet: FC, fecal samples from cows fed the silage diet supplemented with 5% *M. japonica*: FT, rumen samples from cows fed the control silage diet: RC, rumen samples from cows fed the silage diet supplemented with 5% *M. japonica*: RT, environmental samples, which included soil and compost samples as well as filter paper with *M. japonica*: ENV). One soil sample (Sample BM17-E – env_soil_1) mapped significantly fewer proteins in metaproteomic analysis compared to the rest of the data and was removed from further analysis as a technical outlier. Subsequently, this filtering resolved in a total of 4,420 unique protein groups across the 21 samples (Supplementary Data 3). While environmental samples (soil, compost, and filter paper of *M. japonica*) were sequenced and analyzed using the same workflow, their results were omitted from downstream analysis and interpretation of data, as they were outside the primary scope of this study. However, the data are available in the PRIDE repository. For statistical analysis, missing values were imputed by applying a 2.8 downshift from the normal distribution. Protein groups showing significant changes in LFQ intensities ($p < 0.05$) were identified using a two-sided t-test. Dot plots for Fig. 1C was created using ggplot2[66] in R (v. 4.2.2)[79]. Volcano plot for Fig. 1D was created using Perseus (v. 2.0.7.0). Predicted gene organization map was created using the ggplot2 extension package gggenes[80], and protein expression heatmap was created using ggplot2.

### *Bacteroides* culture enrichment and isolation

Rumen samples from cattle in the *ad libitum* feed trial were taken and inoculated directly into anaerobic media within sealed Hungate tubes (atmosphere: 85% $N_2$, 10% $CO_2$, 5% $H_2$). Anaerobic media was composed of *Bacteroides* minimal medium[29] (MM) supplemented with 1% Bacto™ Tryptone (BD), 0.5% Yeast Extract Bacteriological (VWR), and 0.5% *Mj*Ex. Cultures were grown until sufficient density was reached and re-inoculated into fresh media 3X before being glycerol stocked and placed into an anaerobic chamber with similar atmosphere. Cultures were diluted $10^{-3}$ and consecutively streaked on minimalized media agar plates containing 0.5% substrate (κ and ι-carrageenan, and *Mj*Ex) and 1% agar (repeated four to five times) to obtain pure isolates. Isolates were grown in MM + 0.5% galactose.

Further *Bacteroides* enrichments were carried out on fecal samples from Wilder Institute/Calgary Zoo ruminants. Samples were collected from 7 ruminant, 3 pseudoruminant, and 1 nonruminant animals (Supplementary Data 7). All animal diets are included in Supplemental Data 10. 1 g of fecal samples were diluted in 10 mL of anaerobic phosphate saline buffer (PBS) and vortexed to homogenize on-site. 100 μL of homogenized mixture was inoculated in 1:1 anaerobic 2 x MM and *Mj*Ex and incubated for ~48 h.

Isolate DNA was extracted using the Qiagen Powersoil kit and was sequenced using Illumina (NovaSeq 6000 PE 150 bp), and Oxford Nanopore (R9.4.1; SQK-RBK110-96) sequencing. Reads from bacterial isolates from cattle and zoo animal samples were trimmed as above using trimmomatic, and hybrid assembled using Unicyler (v0.4.8)[81]. Isolate genomes were checked for quality using Quast (v5.0.2)[82].

### Carrageenan PUL analysis

Bovine MAGs and ruminant isolates were taxonomically classified using GTDB-Tk (v2.3.2 - R202)[83] and CheckM2 (v1.1.3)[84], and were functionally annotated with DRAM (v1.3.4)[85], prodigal (v2.6.3)[86] and dbCAN3[87]. Protein sequence from CarPULs: *Bx*Car5_BOV CarPUL−1, CarPUL−2, *Bx*MAG_BOV PUL, and *Bx*Car17_BOV were used as a protein database for Diamond BLASTx (v2.1.8)[88] against 17 TB of ruminant metagenomic datasets from cattle, buffalo, deer, moose, yak, and sheep collected from the literature (Supplementary Data 6). Data was collected via NCBI-vdb (v3.1.0). Hits below a Bit score 50 and identity of 50% were filtered out, and the top hit for each SRR was selected for. Buffalo metagenomic datasets were selected for co-assembly with MEGAHIT due to multiple high identity hits against carrageenan PUL sequences (Source Data).

CarPUL CAZyme protein sequences were BLASTed against the NCBI database and *Mj*SM protein database taken in this study. The top 100 hits against CAZymes were collected and ran through SACCHARIS v2 (v2.0.1)[31] with CarPUL sequences. *Rikenellaceae* MAGs were downloaded from BioProject PRJNA1029302, and annotated as above. All phylogenies were visualized and annotated in ITOL[89]. OrthoFinder (v2.5.5)[90] was used to create species trees to align genome PULs in Figs. 2, 4. All PUL diagrams were created using gggenes[80]. All figures were generated in-house using the open-sourced Inkscape editing software (https://inkscape.org).

### *Bacteroides* carrageenan growth profiling

*Bx*Car5_BOV and *Bx*Car17_BOV and wild-type *B. xylanisolvens* XB1A were cultured anaerobically at 37 °C overnight minimalized media + 0.5% galactose overnight. The overnight cultures ($OD_{600\ nm}$ 1.0−1.4) were diluted to an $OD_{600\ nm}$ of 0.025 in 1X MM. Wells of a 96-well microtiter plates (Falcon) were filled with 100 μL inoculant (n = 4) and 100 μL 0.3% (w/v) substrate. *Mj*Ex, κ, λ, and ι-carrageenan were autoclaved, the subsequently dialyzed against 6 kDa to remove any free monosaccharides. Galactose was used as a positive growth control. Negative control wells consisted of 100 μL 2X MM combined with 100 μL 0.3% (w/v) substrate and were used to normalize growth curves. Plates were sealed with polyurethane Breathe-Easy gas-permeable membranes (Sigma; Z390059). $OD_{600\ nm}$ of each well was measured with a Biotek Eon microplate reader and recorded on Biotek Gen5 software every 10 min for 48 h. Mean ( ± standard deviation) of each condition (n = 4) was visualized using GraphPad Prism (v10.2.3).

### *Bx*MAG_BOV GH16_17B production and characterization

*Bx*MAG_BOV GH16_17A & B were codon optimized and synthesized with Biobasic. Expression plasmids were transformed into *Escherichia coli* BL21 (DE3) Tuner competent cells (Novagen) and grown in LB Miller broth containing kanamycin (50 ug mL). Cell cultures were grown to an optical density of 0.6-0.8 at $OD_{600\ nm}$ and induced with isopropyl β-ᴅ-1-thiogalactopyranoside at a final concentration of 1 mM at 16 °C overnight while shaking at 200 rpm. Cells were lysed using a combination of lysis buffer (20 mM Tris pH 8.0, 500 mM NaCl, 0.1 mg mL$^{-1}$ lysozyme) and sonication for 2 min of 1 s intervals of sonic pulses at an intensity amplitude of 30 (Fisherbrand Model 705 Sonic Dismembrator and probe; Thermo Fisher Scientific). Cell lysate was centrifuged at $17,500 \times g$ for 45 min and purified using Ni-NTA resin columns (Cytiva) and immobilized metal affinity chromatography. Protein was eluted by an increasing gradient 0−500 mM imidazole in 20 mM Tris pH 8.0 and 500 mM NaCl. Samples were buffer exchange with 20 mM Tris, 500 mM NaCl, and 2% glycerol using ultrafiltration column, and further purification by size exclusion chromatography HiPrep 16/60 Sephacryl S-200 HR column (GE Healthcare). All fractions were confirmed using sodium dodecyl sulfate−polyacrylamide gel electrophoresis.

Optimum pH was determined using a copper bicinchoninic acid (BCA) reducing sugar assay measured with a SpectraMax ID3 (Molecular Devices) at 565 nm. Briefly, reducing sugar was measured from the release of substrate (3 mg mL$^{-1}$ k-carrageenan) in the presence of 1 μM enzyme and 15 mM buffer. (citrate phosphate pH 3-8; bicine 8-9, CHES 10) after 5 minutes. Activity was plotted using GraphPad Prism (v10.2.3). Future reaction digests were completed at optimum pH. GH16 digests of κ, λ, and ι-carrageenan (carbosynth), and *Mj*Ex was conducted overnight with 3 mg mL$^{-1}$ substrate, and were boiled at 90 °C for 10 min to complete the reaction. Digests were separated using thin layer chromatography on silica gel plates (Millipore) in a 1-butanol: distilled water: acetic acid (2:1:1 v/v/v)

mobile phase and visualized with an ethanol: sulfuric acid (70:3 v/v) solution containing 1% (w/v) orcinol monohydrate (Sigma) and heated at 105 °C for 3 min. 1 mM κ -carrageenan standards (Dextra) were used as reference.

Carrageenan digest were repeated at 5 mg mL$^{-1}$ and dialyzed in 2 kDa dialysis cutoff tubing against ultrapure 18 MΩ cm-1 water, retaining the dialysate. After 3 water changes, dialysate was frozen and lyophilized to dryness. Samples were then resuspended in ultrapure water. Liquid chromatography with electrospray ionization mass spectrometry (LC-ESI-MS) was performed on a Vanquish ultra-high performance liquid chromatography (UHPLC) system (Thermo Scientific). Separation of the carrageenan oligosaccharides was achieved using an Acquity UPLC BEH Amide (HILIC) Column, 130 Å, 1.7 μm, 2.1 mm × 150 mm (Waters) at a flow rate of 300 μL min$^{-1}$ at 30 °C, using a gradient over 20 min from 5-60% 10 mM ammonium formate pH 4.5 and 95-40% 10 mM ammonium formate pH 4.5 in acetonitrile.

Carrageenan oligosaccharide samples were prepared in water and injected in a volume of 10 μL at a concentration of 200 μg mL$^{-1}$ for electrospray ionization mass spectrometry (ESI-MS) on an Orbitrap Fusion Tribrid system (Thermo Scientific) in negative ion mode. Mass spectra parameters are shown in (Supplementary Table 2). To select ions for MS2 experiments, an intensity threshold filter was employed with a minimum intensity of 25,000 and maximum intensity of 1E + 20, and a dynamic exclusion filter was used after 1 times for 2.5 s with default mass tolerance values. Higher-energy collisional dissociation (HCD) was employed to generate fragments. MS spectra were analyzed using Xcalibur and Freestyle software packages (Thermo Scientific). Samples were run multiple times during optimization strategies with the final run appearing in the manuscript.

### $Bx$MAG$_{BOV}$ GH16_17A protein production and purification

The $Pf$S1_19B sulfatase encoding plasmid was co-transformed with pBAD/myc-his A Rv0712 (FGE) (Addgene plasmid no. 16132) into $E. coli$ BL21 (DE3) Star cells (Invitrogen) and grown in LB medium containing 50 μg mL$^{-1}$ kanamycin sulfate, 100 μg mL$^{-1}$ ampicillin and 50 μg mL$^{-1}$ chloramphenicol. Cells were grown at 37 °C until the cell density reached an OD$_{600}$ nm of roughly 0.5, at which time the temperature was dropped to 16 °C and FGE expression was induced with 0.02% L-arabinose. After roughly 2 h, sulfatase expression was induced with a final concentration of 0.5 mM IPTG and the culture incubated for a further 16 h. $Bx$MAG$_{BOV}$ GH16_17A and $Pf$GH16B for enzyme assays and crystallography were produced as follows. A preculture of BL21 (DE3) transformed with pET28 expression plasmids-$Bx$MAG$_{BOV}$ GH16_17A was produced as follows. A preculture of BL21/DE3 transformed with pET28-$Bx$MAG$_{BOV}$ GH16_17A was prepared by combining 50 μL of thawed glycerol stock with 40 mL of LB media containing the construct-specific antibiotics. The preculture was incubated overnight at 37 °C with shaking (170 rpm). The preculture was added to 2 L of 2X YT media containing the same construct-specific antibiotics and any necessary cofactors. The mixture was incubated at 37 °C with shaking until OD$_{600}$ nm ~ 0.6 was reached. The cell mixture was cooled to 16 °C, expression of the construct was induced, and incubated overnight at 16 °C with shaking. The cell mixture was centrifuged at 5400 × $g$ for 15 min at 4 °C. The pellet was resuspended in 10 mL of 25% sucrose and lysed by adding 10 mg of lysozyme, which was left to stir for 20 min. The lysis mixture was combined with 30 mL of deoxycholate and was left to stir for 10 minutes. The lysed cells were centrifuged at 5400 × $g$ for 60 min at 4 °C. $Bx$MAG$_{BOV}$ GH16_17A was purified from clarified cell lysate via immobilized metal affinity chromatography using solutions buffered with 20 mM TRIS, pH 8.0, containing 100 mM NaCl (Binding buffer). Washes were performed with binding buffer containing 10, 20, then 30 mM imidazole. Elution was performed with binding buffer containing 500 mM imidazole. Imidazole used for elution was removed by dialysis into 20 mM TRIS pH 8.0, 100 mM NaCl overnight

at 4 °C. Protein was concentrated using a stirred ultrafiltration cell with a 10 kDa molecular weight cutoff filter. $Bx$MAG$_{BOV}$ GH16_17A was further purified by gel filtration chromatography using an S100 Sephacryl column and 20 mM TRIS, pH 8.0, 100 mM NaCl. Protein was again concentrated by ultrafiltration and quantified by absorbance at 280 nm.

### $Bx$MAG$_{BOV}$ GH16_17A enzyme assays

Prior to performing the enzymes assays, a 1% (w/v) solution of ι-carrageenan was prepared in 20 mM TRIS, pH 8.0, containing 100 mM NaCl. A 5 mL aliquot was taken, placed into a 15 mL falcon tube, and incubated with 5 mM PfS1_19B for 7 days with shaking at room temperature. A matched control with no PfS1_19B was also incubated for 7 days. Both solutions were centrifuges for 3 min at 5400 × $g$, the supernatant was collected, then filtered through a sterile syringe filter (0.22 μM). The untreated and PfS1_19B treated carrageenan was then used for digests with $Pf$GH16B and $Bx$MAG$_{BOV}$ GH16_17A, both at 5 μM. Reactions were incubated for 3 days with shaking at room temperature.

Samples were labeled for fluorophore-assisted carbohydrate electrophoresis (FACE). Each digestion reaction was killed by adding 1 mL 95% EtOH and dried for 3 hours in the speed vacuum. Samples were resuspended in 5 μL of each 0.02 M ANTS (8-aminonaphthalene-1,3,6- trisulfonic acid) and 0.1 M NaCNBH$_3$, vortexed, spun down for 30 s at 5,000 rpm, and was left to incubate overnight at 37 °C wrapped in foil. Samples were dried for 4 hours in the speed vacuum. Once dried, each sample was resuspended in 500 μL of FACE loading dye and 5 μL of each sample was loaded into a FACE gel. The FACE gel was composed of a 35% acrylamide resolving gel and a 10% acrylamide stacking gel. The gel was visualized using a MultiImage™ Light Cabinet and AlphaImager software.

### $Bx$MAG$_{BOV}$ GH16_17A crystallography

Crystals were grown at 18 °C using sitting-drop vapor diffusion for screening and hanging drop vapor diffusion for optimization. $Bx$MAG$_{BOV}$ GH16_17A at 5 mg ml- in 20 mM Tris-HCl, pH 8.0, with 0.5 M NaCl was crystallized in a 1:1 ratio with a solution of 0.2 M NaCl, 0.1 M Na$_2$HPO$_4$:Citric acid, pH 4.2, and 20% PEG8000. A single crystal of $Bx$MAG$_{BOV}$ GH16_17A in crystallization solution supplemented with 20% (v/v) ethylene glycol as cryoprotectant was flash cooled by looping and plunging into liquid nitrogen. Diffraction data was collected on CMCF-ID beamline at the Canadian Light Source (CLS, Saskatoon, Saskatchewan) as indicated in (Supplementary Table 3). All diffraction data were processed using XDS[91] and Aimless[92]. Data collection and processing statistics are shown in (Supplementary Table 3). Initial phases were determined by molecular replacement using PHASER[93] and a model of $Bx$MAG$_{BOV}$ GH16_17A generated with AlphaFold 2[94]. This initial model was manually corrected with COOT[95] and refinement of atomic coordinates was performed with Phenix.refine[96]. Water molecules were added in COOT with FIND-WATERS and manually checked after refinement. In all datasets, refinement procedures were monitored by flagging 5% of all observations as "free"[97]. Model validation was performed with MOLPROBITY[98].

### Production of fluorescent polysaccharides

Fluorescently labeled $Mj$Ex (FLA-$Mj$Ex) and amylopectin (FLA-AP) were produced following a described protocol[96] with slight modifications. Prior to labeling, $Mj$Ex and AP were hydrolyzed by mild enzyme hydrolysis. Autoclaved $Mj$Ex (10 mg mL$^{-1}$) was treated with 2 μM $Bx$MAG$_{BOV}$ GH16B in 0.02 M citrate phosphate buffer, pH 5.5 for 3 h at 37 °C with mild shaking (50 rpm). The enzyme was then heat killed by boiling the solution in a hot water bath for 10 minutes. The sample was centrifuged and supernatant collected to run through a 5 kDa MWCO Vivaspin column (Sartorius) to remove low MW products. AP

from maize (Sigma) was lightly digested with alpha-amylase from *Bacillus licheniformis* (Sigma) for 30 min, before heat killing and spinning out the enzyme. Low molecular weight products were removed using a Vivaspin column. The digested products were activated using CNBr and cleaned up with a Sephadex® G-50 gel filtration medium in a column connected to an ÄKTA Start™ chromatography system before incubating with fluoresceinamine isomer II (FLA; Sigma) overnight. Excess FLA was removed using a Vivaspin column as above until filtrate was clear. Retentate was collected, lyophilized, and stored at 4 °C.

### Fluorescent polysaccharide incubations and imaging

$Bx$Car5$_{BOV}$, $Bx$Car17$_{BOV}$, and $Bx$XB1A$_{HOM}$ glycerol stocks were used to inoculate 0.5% Gal-MM and cultures were grown anaerobically. Overnight cultures were used to inoculate Gal-MM (not primed) or $Mj$Ex-MM (primed). Instead of $Mj$Ex-MM, $Bx$XB1A was subcultured into 50:50 Gal:$Mj$Ex -MM due to lack of grown on $Mj$Ex as the sole carbon source. Cultures were centrifuged for 5 min at $5000 \times g$ and pellets were resuspended in 2X MM. All cultures were diluted to 0.8 OD$_{600}$ nm. For 0 h controls, 50 μL of resuspended cultures were aliquoted into 500 μL 1% formaldehyde (FA). 20 μL of resuspended cultures were then subcultured into tubes containing 50 μL 2X MM and 50 μL 0.4% FLA-$Mj$Ex, FLA-AP, $Mj$Ex, AP, or FLA + $Mj$Ex. 50 μL of culture was aliquoted into 500 μL of FA at 1 h and 1 d. Samples were fixed in FA overnight at 4 °C before centrifuging ($5000 \times g$, 10 min), removing the supernatant, and resuspending in 500 μL 1X PBS. Cells were imaged as previously described[97]. Briefly, samples were diluted 1 in 5 with 1X PBS buffer and fixed onto an Isopore™ filter using gentle vacuum. A 4:1 Citifluor™ AFI mountant medium to Vectashield™ vibrance antifade mounting medium solution containing 4′6-diamidino−2-phenylindole (DAPI) was used to mount and counterstain the filter pieces. A Leica DMRBE epifluorescence microscope coupled to a Leica DFC 7000 T camera and a X-cite 110 LED illumination system was used to image the samples. DAPI and FITC filter cubes were used during imaging.

Microscopy was conducted with an ECHO Microscope RVL2-K4 (A BICO COMPANY, USA) equipped with LED light cubes DAPI (EX: 385/30 EM: 450/50 DM: 425) and FITC (EX: 470/40 EM: 525/50 DM: 495) using a Plan X Apo oil; 1,42 NA, 60× oil emersion objective. For the FLA-$Mj$Ex imaging fixed LED strength of 46% and fixed exposure times of 40 ms were applied. The threshold of 40 ms was applied to prevent false positive analysis of cell autofluorescence. The images were processed using the Echo Pro software version 6.4.2 (Echo Laboratories). For the quantification of total FLA-$Mj$Ex manual counting was performed of a minimum of 15 fields of view from each sampling time point. Positive FLA-$Mj$Ex cells were identified by a co-localized DAPI and FLA-$Mj$Ex signal.

For Super Resolution Structured Illumination (SR-SIM) microscopy, the cells were filtered onto a 0.2 μm pore size polycarbonate filter using a gentle vacuum of 200 mbar. The filters were mounted in a Citifluor/VectaShield (4:1) mounting solution containing 1 ng μL$^{-1}$ DAPI. FLA-stained cells were visualized and enumerated using a fully automated microscope imaging system[99] on a Zeiss AxioImager.Z2 microscope stand equipped with a cooled charged-coupled-device camera (AxioCam MRm + Colibri LED light source), three light-emitting diodes (UV-emitting LED at 365 nm for DAPI and blue-emitting LED at 470 nm for FLAPS 488), and a HE-62 multi-filter module with a triple emission filter (425/50 nm, 527/54 nm, LP 615 nm) incorporating a triple beam splitter of 395/495/610, microscopy was conducted using a 63× magnification oil immersion plan apochromatic objective with a numerical aperture of 1.4 (Carl Zeiss). All automatic cell counts were validated using manual cell counting. Briefly automated cell counting was carried out by initially acquiring images, at selected wavelengths (DAPI, FLAPS), of a previously defined set of coordinates consisting of a minimum of 46 fields of view on each sample filter[99]. Subsequently, the

images were imported into the ACMETOOL3.0 (http://www.technobiology.ch and Max Planck Institute for Marine Microbiology, Bremen) image analysis software. From the images, cells were deemed 'substrate stained' if they showed a positive signal in both the DAPI and FLAPS (488) images. Additionally, these signals had to have a minimum overlap of 30%, a minimum area of 20 pixel (0.17−0.3 μm²) (DAPI signal and FLAPS signal, respectively) and a minimum signal background ratio of 1.

### Statistics & reproducibility

All descriptions are provided in their associated methods or in the source data files. No statistical method was used to predetermine sample sizes. No data were excluded from the analyses. The experiments were not randomized; the investigators were not blinded to allocation during experiments and outcome assessment, but a randomized scheme was applied during omics sample preparation. Bioinformatic analysis was same for all diet groups and processed together under identical conditions.

### Reporting summary

Further information on research design is available in the Nature Portfolio Reporting Summary linked to this article.

### Data availability

The mass spectrometry proteomics data have been deposited to the ProteomeXchange Consortium via the PRIDE partner repository with the dataset identifier PXD060679. All sequencing data have been deposited to the NCBI sequence read archive (SRA) under the project accession numbers PRJNA1227608. The functionally annotated MAGs and the $Mj$SM protein database are available on Figshare (https://doi.org/10.6084/m9.figshare.28464551). The coordinates and Crystal structure factors for $Bx$MAG$_{BOV}$ GH16_17A have been deposited in submitted to the Protein Data Bank under the identifier 9EFL. LC-ESI-MS data is deposited on GlycoPOST under the ID GPST000613. Previously published datasets used in this study as reference material are provided in the Source Data. Source data are provided with this paper.

### Code availability

R scripts used for 16S analysis can be found on Zenodo (https://doi.org/10.5281/zenodo.18775548).

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

## Acknowledgements

Research at AAFC was supported through an AgriScience Project (ASP–207; J-002817; D.W.A.) and Prize for Outstanding Achievement in Science (AAFC) (J-003135; D.W.A.). L.H.H. is grateful for support from The Research Council of Norway (302639 – SeaCow; L.H.H.). P.B.B. is grateful for support from the Novo Nordisk Foundation (0054575—SuPAcow; P.B.B.) and the Australian Research Council (Future Fellowship: FT230100560; P.B.B.). Research at the University of Victoria was supported by a Natural Sciences and Engineering Research Council of Canada Discovery Grant (FRN 04355; A.B.B.). T.R.P. was supported by the Canada Research Chair program (CRC-2021-00420; T.R.P). G.R. was supported through funding from the Deutsche Forschungsgemeinschaft (DFG, German Research Foundation; project number 496342779; G.R.). Mass spectrometry-based proteomic analyses were performed by The MS and Proteomics Core Facility, Norwegian University of Life Sciences (NMBU). This facility is a member of the National Network of Advanced Proteomics Infrastructure (NAPI), which is funded by the Research Council of Norway INFRASTRUKTUR-program (project number: 295910). We acknowledge the assistance of Dr. John Church (Thompson Rivers University) in assisting to organize the farm trial. We thank Arthur Valdivieso for technical assistance with *Bx*MAG$_{BOV}$ GH16_17A activity experiments. We are thankful for the generous access provided by the Molecular Ecology Department of the Max Planck Institute for Marine Microbiology. We also greatly appreciate the Animal Care, Health and Welfare staff at the Wilder Institute/Calgary Zoo for facilitating access to and collecting feces for the study. LC-ESI-MS data was collected at the University of Lethbridge, Lethbridge Central Analytical Facility, with support from Tony Montina, Vincent Weiler, Maurice Needham, and Carl Holland. Insightful bioinformatic suggestions were provided by Rodrigo Ortega Polo at Agriculture & Agri-Food Canada Lethbridge.

## Author contributions

D.W.A and J.P.T conceived and designed the study. T.O.A, I.A., L.H.H., and P.B.P. conducted proteomic analysis. X.X. performed linkage analysis on *M. japonica*. J.P.T., G.R., L.K., A.Y.S., S.S., E.S., and D.P.W. collected and enriched samples for isolation and metagenomics. J.P.T. and G.R. conducted 16S analysis. J.P.T performed shotgun metagenomic analyses and metagenomic surveys. J.P.T performed growth curves and isolations. J.P.T, L.M., K.E.L., and N.J. designed GH16 constructs, purified protein, and characterized CAZymes. L.M. and A.B.B. conducted crystallography. G.R., L.K., and A.Y.S conducted FISH and FLA-*Mj*Ex on enrichments and isolates. J.P.T., T.O.A., D.W.A., and P.B.P. drafted the manuscript. D.W.A, P.B.P, and T.R.P. supervised the work. All authors wrote, edited, and approved the manuscript.

## Competing interests

The authors declare no competing interests.

## Additional information

¹Lethbridge Research and Development Centre, Agriculture and Agri-Food Canada, Lethbridge, AB, Canada. ²Department of Chemistry and Biochemistry, University of Lethbridge, Lethbridge, AB, Canada. ³Department of Animal and Aquacultural Sciences, Faculty of Biosciences, Norwegian University of Life Sciences, Ås, Norway. ⁴Faculty of Chemistry, Biotechnology and Food Science, Norwegian University of Life Sciences, Ås, Norway. ⁵Department of Biochemistry and Microbiology, University of Victoria, Victoria, BC, Canada. ⁶Faculty of Veterinary Medicine, University of Calgary, Calgary, Canada. ⁷Animal Health Department, Wilder Institute/Calgary Zoo, Calgary, Canada. ⁸MACE Laboratory, ALPOLE-ENAC, École Polytechnique Fédérale de Lausanne, Sion, Switzerland. ⁹Spoitz Enterprises Inc, Courtenay, BC, Canada. ¹⁰Beaver Meadow Farms, Comox, BC, Canada. ¹¹Microbial-Carbohydrate Interactions

Group, Faculty 2 Biology / Chemistry, University of Bremen, Bremen, Germany. [12]Alberta RNA Research and Training Institute and Department of Chemistry and Biochemistry, Lethbridge, AB, Canada. [13]Department of Microbiology, Immunology and Infectious Disease, Cumming School of Medicine, University of Calgary, Calgary, AB, Canada. [14]Centre for Microbiome Research, School of Biomedical Sciences, Queensland University of Technology (QUT), Translational Research Institute, Woolloongabba, QLD, Australia. ✉e-mail: wade.abbott@agr.gc.ca

