## [Transparent Peer Review file · Nature Communications]

Distribution of microbial carrageenan foraging pathways reveals a widespread latent trait within the ruminant intestinal microbiome

Corresponding Author: Dr Wade Abbott

Version 0:

Reviewer comments:

Reviewer #1

(Remarks to the Author)

In this manuscript, Tingley and al, investigate the potential of *Mazzaella japonica* as a sustainable feed for livestock, focusing on its impact on ruminant microbiomes, particularly regarding carrageenan digestion. The authors report increased *Bacteroides* abundance in the distal gut and identify carrageenan-active polysaccharide utilization loci (CarPULs). Carrageenan-active polysaccharide utilization loci (CarPULs) were identified and characterized in the study, revealing that carrageenan catabolism is widespread among ruminants globally. The authors claim these pathways are structurally distinct from those in marine bacteria, and suggest a complex and ancient evolutionary history in ruminant microbiomes.

The genomic and biochemical data presented in the manuscript are well-executed and provide valuable insights. However, there are some gaps in the evidence regarding the impact of *M. japonica* on the rumen and faecal microbiomes, particularly concerning the claims about *Bacteroides* abundance. Additional data, such as metagenomic sequencing and proteomics of the algae (which the authors have already performed but not presented effectively) is essential to strengthen the manuscript and provide a clearer understanding of the findings and their novelty.

1. As stated above, the major issue with the manuscript is the lack of clarity about whether *Mazzaella japonica* affects the rumen microbiome or if it passes undigested and is excreted in the faeces. Including metagenomic sequencing and proteomics data for the *Mazzaella japonica* surface-associated microbiome (MjSM) in main figures would allow for a thorough comparative analysis of its digestive fate and microbial effects and help assess its true impact on the ruminant microbiome.
2. The manuscript claims *M. japonica* increases *Bacteroides* abundance in the distal gut compared to the rumen. However, the distal gut was not sequenced, making this claim unsupported. The authors should reconsider or rephrase this claim based on available data.
3. More background information on *Mazzaella japonica* is needed to justify its use in livestock feeding. The manuscript should provide more details on its potential as a methane inhibitor, its widespread use in agriculture, and evidence supporting its use as a feed supplement.
4. The manuscript contains several unexplained abbreviations, such as "BxB1AHOM," which could refer to a control strain. Be careful that all abbreviations are clearly defined on their first. These clarifications would improve readability of the manuscript.
5. The manuscript suggests that sequencing depth was insufficient to detect low-abundance PULs and CAZymes. Including rarefaction curves would support this claim and help assess the validity of the authors' conclusions regarding the detection of these microbial pathways.

(Remarks on code availability)

Reviewer #2

(Remarks to the Author)

(Remarks on code availability)

The study provides a comprehensive analysis of carrageenan degradation in cattle, combining 16S rRNA sequencing, metagenomics, metaproteomics, and biochemical characterization. The identification of CarPULs in bovine microbiomes and their homology to marine and other mammalian microbiomes is novel and contributes significantly to understanding microbial adaptation to dietary polysaccharides in ruminants. Although the authors have characterized the metabolism of *M. japonicum* extracts via CarPULs, most of the findings depicted in here are association studies. Here are some concerns related to the study:

Minor corrections/suggestions involve

1. The authors showed conserved CarPULs in *Bacteriodes* species, however, they have not demonstrated that lack of gene, renders the bacteria incapable to grow in seaweed extracts. The authors need to perform knockout or knockdown studies to show direct correlation between the enzyme and Carrageenan degradation as the loci has other genes as well that are hypothetical.
2. Another drawback of this study is that the association of bacteriodes with *M. japonicum* diet has been only shown in one cohort of cattle. The authors should discuss the limitations of this in the results or discussion section.
3. The title needs to be rephrased. Instead of latent traits – mention what traits; instead of genetic “dark matter” mention what genetic elements they have identified. Same goes with the abstract as it vaguely mentions that they “characterized the gene to provide insights”. The authors should mention specifically (in general terminology) what characterization they have done and how it is novel.
4. The authors provide limited details on the recombinant protein purification protocol, such as buffer compositions and elution gradients used. Could the authors include a more comprehensive description of these methods in the main text or supplementary materials to enhance reproducibility?
5. Since part of the study involves characterization of the enzyme, could the authors provide an SDS-PAGE gel image of the purification, gel filtration chromatogram to determine the stoichiometry of the enzyme. It is important to confirm the purity of the purified protein(s) as well as their oligomeric state when used for assays, such as for BxMAGBOV GH16_17A or GH16_17B? This would strengthen the reliability of the enzymatic data.
6. The authors demonstrate phylogenetic variability in GH16_17 genes across diverse species (Fig. 3A). Could the authors further investigate whether the active site architecture is conserved across these variants by modeling the active sites of representative GH16_17 enzymes to highlight key binding site residues?
7. The authors selected BxCar5BOV and BxCar17BOV for long-read genome sequencing (Lines 149–150) but focused biochemical characterization on BxMAGBOV GH16_17A and GH16_17B. Could the authors clarify why they were not shortlisted for characterization?
8. The E280 residue in BxMAGBOV GH16_17A is critical for substrate specificity (Line 190). Could the authors confirm whether this residue is conserved in the corresponding position of GH16_17 enzymes from BxCar5BOV (CarPUL-1 or CarPUL-2) and BxCar17BOV, using sequence alignments or structural modeling?
9. In Fig. 3B–C, the structural figures display all active site residues as stick models, which may obscure key features. Could the authors simplify these panels by highlighting only the critical E280/G258 residue and the substrate to improve clarity and emphasize the structural determinant of substrate specificity?
10. The authors note that BxCar17BOV has one CarPUL, while BxCar5BOV has two, with CarPUL-1 sharing 99–100% identity with BxMAGBOV's CarPUL (Lines 162–164). Could the authors provide the sequence identity between the GH16_17 enzyme in BxCar17BOV's CarPUL and BxMAGBOV's GH16_17 enzymes? Additionally, do the GH16_17 enzymes from BxCar17BOV, BxCar5BOV (CarPUL-1 and CarPUL-2), and BxMAGBOV share similar active site architectures, as determined by sequence alignment or structural modeling?
11. Line 458, it should be kappa carrageen's, not “k”
12. The data refinement statistics are missing the cell dimensions. Please incorporate that.
13. Can the authors provide the PDB validation report for reviewing?

Reviewer #3

(Remarks to the Author)

(Remarks on code availability)

Version 1:

Reviewer comments:

Reviewer #1

(Remarks to the Author)

Original comment:

Including metagenomic sequencing and proteomics data for the *Mazzaella japonica* surface associated microbiome (MjSM) in main figures would allow for a thorough comparative analysis of its digestive fate and microbial effects and help assess its true impact on the ruminant microbiome.

Response: We thank the reviewer for their suggestion; however, we believe there may be some confusion on how the MjSM microbiome was studied and its contributions to this study. The MjSM is the microbiota scraped from the surface of *M. japonica* collected directly sourced from the beach. This community would not appear in abundance within the faecal or rumen metagenomics/metaproteomics nor would it provide insight on the ruminant digestion of *M. japonica* as it would not be viable in the digestive organs of the animal. We have made a more detailed note of this in the methods: Lines 719-723: "While environmental samples (soil, compost, and filter paper of *M. japonica*) were sequenced and analyzed using the same workflow, their results were omitted from downstream analysis and interpretation of data, as they were outside the primary scope of this study. However, the data are available in the PRIDE repository." This dataset was intended to provide an inventory of genes present on the marine environment microbiome to explore putative horizontal gene transfer events within the rumen community none of which was detected. Hence the inclusion of the MjSM microbiome was only included in the supplemental material as a negative result. We have changed the acronym to "surfaceassociated" microbiota for clarity (Line 262).

Reviewer response: I thank the authors for their response, but I remain concerned that omitting the *Mazzaella japonica* surface-associated microbiome (MjSM) from the analyses and manuscript leaves a critical gap. In particular, the *Bacteroides* isolates and associated enzymatic functions reported in this study may in fact derive from the MjSM rather than being intrinsic to the rumen or faecal microbiome. Without a direct comparative analysis, it is difficult to exclude this possibility. I therefore strongly encourage the authors to include at least a summary comparison between MjSM and the rumen/faecal datasets, highlighting whether or not *Bacteroides* species and associated carbohydrate-active enzymes overlap. This would not only address the concern but also provide transparency and accessibility of the results. Also the pride repository and figshare were not accessible for the reviewers.

My other remarks were addressed adequately by the authors.

Reviewer #2

(Remarks to the Author)

Thank you for modifying and addressing the comments.

Regarding response to Q2.1

While the authors correctly acknowledge that generating knockouts in environmental *Bacteroides* isolates is technically challenging, the current evidence remains strictly correlative rather than causative. Without genetic validation, the functional link between the CarPUL and carrageenan metabolism cannot be definitively established.

A gain-of-function experiment would provide the most direct and interpretable validation. For example, introducing the CarPUL locus (e.g., from BxMAGBOV) into a genetically tractable host that lacks the native pathway, such as *Escherichia coli* or a laboratory *Bacteroides* strain, and subsequently testing for the ability to degrade *M. japonica* extracts using established assays (e.g., FACE or LC-MS; see Lines 201–211), would demonstrate sufficiency of the CarPUL for carrageenan utilization. Even a partial reconstruction or expression of key CarPUL enzymes in such hosts would meaningfully strengthen the mechanistic argument.

In addition to traditional mutagenesis, recent advances in *Bacteroides* genetics now enable CRISPR interference (CRISPRi)-based knockdown of individual genes without requiring double recombination or strain engineering. CRISPRi systems employing catalytically inactive Cas9 (dCas9) have been successfully adapted for *Bacteroides* (<https://pmc.ncbi.nlm.nih.gov/articles/PMC10861873/>). Even if environmental isolates are difficult to transform, such tools could be implemented in a CRISPRi-compatible model strain background to verify the contribution of specific CarPUL genes to carrageenan degradation. Attempting such an experiment, or at least discussing why it could not be done, would demonstrate due diligence.

If genetic perturbation truly remains infeasible (after attempting the experiment), the authors should explicitly acknowledge this limitation in the Discussion, stating that the findings establish associative evidence but stop short of demonstrating causality. Merely correlating the presence of CarPULs with carrageenan utilization, without any perturbation-based validation (knockout, knockdown, or complementation), does not meet the evidence required to infer functionality.

Reviewer #3

(Remarks to the Author)

Version 2:

Reviewer comments:

Reviewer #1

(Remarks to the Author)

The authors addressed my comments.

Reviewer #1 (Remarks to the Author):

In this manuscript, Tingley and al, investigate the potential of *Mazzaella japonica* as a sustainable feed for livestock, focusing on its impact on ruminant microbiomes, particularly regarding carrageenan digestion. The authors report increased *Bacteroides* abundance in the distal gut and identify carrageenan-active polysaccharide utilization loci (CarPULs). Carrageenan-active polysaccharide utilization loci (CarPULs) were identified and characterized in the study, revealing that carrageenan catabolism is widespread among ruminants globally. The authors claim these pathways are structurally distinct from those in marine bacteria, and suggest a complex and ancient evolutionary history in ruminant microbiomes.

The genomic and biochemical data presented in the manuscript are well-executed and provide valuable insights. However, there are some gaps in the evidence regarding the impact of *M. japonica* on the rumen and faecal microbiomes, particularly concerning the claims about *Bacteroides* abundance. Additional data, such as metagenomic sequencing and proteomics of the algae (which the authors have already performed but not presented effectively) is essential to strengthen the manuscript and provide a clearer understanding of the findings and their novelty.

Q1.1. As stated above, the major issue with the manuscript is the lack of clarity about whether *Mazzaella japonica* affects the rumen microbiome or if it passes undigested and is excreted in the faeces.

Response: We thank the review for their concern and agree that this topic was not fully addressed within the main text of the manuscript. The digestibility of *Mazzaella japonica* was reported in an earlier publication: Terry et al., 2023 (DOI: 10.3389/fanim.2023.1181768), which concluded that *M. japonica* is digested within cattle and meets dietary requirements for cattle when supplemented in the diet at 2%. To improve our manuscript we have moved supplemental note 1 into the main text discussion and described the findings of Terry et al. (2023) on lines 279-298.

Including metagenomic sequencing and proteomics data for the *Mazzaella japonica* surface-associated microbiome (MjSM) in main figures would allow for a thorough comparative analysis of its digestive fate and microbial effects and help assess its true impact on the ruminant microbiome.

Response: We thank the reviewer for their suggestion; however, we believe there may be some confusion on how the MjSM microbiome was studied and its contributions to this study. The MjSM is the microbiota scraped from the surface of *M. japonica* collected directly sourced from the beach. This community would not appear in abundance within the faecal or rumen metagenomics/metaproteomics nor would it provide insight on the ruminant digestion of *M. japonica* as it would not be viable in the digestive organs of the animal. We have made a more detailed note of this in the methods:

Lines 719-723: "While environmental samples (soil, compost, and filter paper of *M. japonica*) were sequenced and analyzed using the same workflow, their results were omitted from downstream analysis and interpretation of data, as they were outside the primary scope of this study. However, the data are available in the PRIDE repository."

This dataset was intended to provide an inventory of genes present on the marine environment microbiome to explore putative horizontal gene transfer events within the rumen community - none of which was detected. Hence the inclusion of the *MjSM* microbiome was only included in the supplemental material as a negative result. We have changed the acronym to “surface-associated” microbiota for clarity (Line 262).

Q1.2. The manuscript claims *M. japonica* increases *Bacteroides* abundance in the distal gut compared to the rumen. However, the distal gut was not sequenced, making this claim unsupported. The authors should reconsider or rephrase this claim based on available data.

Response: We agree that directly associating the faecal microbiome with the distal gut microbiome can be misleading (however, it is a common surrogate dataset described in the literature). The following corrections have been made:

Line 41: distal gut replaced with faeces

Line 98: distal gut replaced with faecal

Q1.3. More background information on *Mazzaella japonica* is needed to justify its use in livestock feeding. The manuscript should provide more details on its potential as a methane inhibitor, its widespread use in agriculture, and evidence supporting its use as a feed supplement.

Response: Please see the response to Q1.1. Terry et al., 2023 (10.3389/fanim.2023.1181768) concluded that although *M. japonica* is an acceptable feed supplement in cattle, it is likely not a means to reduce methane emissions in cattle. In addition, its impact as an enteric methane inhibitor and its (negative) impact on the rumen microbiome was discussed in detail in the previous publication: O'Hara et al, 2023 (10.3389/fmicb.2023.1104667). We have refined our discussion on this topic:

Lines 296-298: “It should be noted that *M. japonica* supplementation was shown to be digestible in ruminants, meeting nutritional crude protein requirements; however, not significantly reducing methane emissions.”

Q1.4. The manuscript contains several unexplained abbreviations, such as “BxXB1AHOM,” which could refer to a control strain. Be careful that all abbreviations are clearly defined on their first. These clarifications would improve readability of the manuscript.

Response: We understand that there are many abbreviations in the manuscript and the confusion with this callout in particular. We have addressed this in lines 162-163. We have double-checked that each acronym is defined at first use.

Q1.5. The manuscript suggests that sequencing depth was insufficient to detect low-abundance PULs and CAZymes. Including rarefaction curves would support this claim and help assess the validity of the authors' conclusions regarding the detection of these microbial pathways.

Response: Unfortunately, rarefaction curves are not always acceptable for shotgun metagenomics, and often discouraged <https://doi.org/10.1016/j.cell.2016.08.007> as background functional diversity is already high. Therefore, the identification of observed species, whether through 16S rRNA sequencing or shotgun metagenomics, is not a sufficient method for

uncovering low abundance functional pathways. Because of these reasons we chose not include this analysis in the manuscript. However, to address the reviewer's concern we have attached a 16S rRNA rarefaction curve of the silage and *ad libitum* cattle trials. These results demonstrated no correlation in the between enriched, and non-enriched genomes. Increased abundance overserved in Fig. 3 results in lower diversity and higher background of abundant species. Therefore, we concluded that rarefaction analysis is not sufficient to detect low abundance strains.

Reviewer #2 (Remarks on code availability):

The study provides a comprehensive analysis of carrageenan degradation in cattle, combining 16S rRNA sequencing, metagenomics, metaproteomics, and biochemical characterization. The identification of CarPULs in bovine microbiomes and their homology to marine and other mammalian microbiomes is novel and contributes significantly to understanding microbial adaptation to dietary polysaccharides in ruminants. Although the authors have characterized the metabolism of *M. japonicum* extracts via CarPULs, most of the findings depicted in here are association studies. Here are some concerns related to the study:

Minor corrections/suggestions involve

Q2.1. The authors showed conserved CarPULs in *Bacteroides* species, however, they have not demonstrated that lack of gene, renders the bacteria incapable to grow in seaweed extracts. The authors need to perform knockout or knockdown studies to show direct correlation between the enzyme and Carrageenan degradation as the loci has other genes as well that are hypothetical.

Response: Of note, the bacteria described in this study are environmental isolates, not lab strains, and no genetic tools exist to mutate them. Environmental strains are notorious for being genetically immutable. In a related *Bacteroides* paper (10.1038/s41467-025-55845-7), the authors state: "environmental bacteria were not feasible for engineering." Also, knockdowns, which also require molecular transformation and tools for genetic manipulation of bacteria, are

not suitable to test pathways such as this that contain functional redundancy. For example, *Bacteroides* has been shown to grow after RNAi treatment (10.1038/s41467-024-48802-3). Therefore, to provide a negative control, we selected a closely related *Bacteroides xylanosolvens* strain isolate that was available within a public repository (BxXB1) that lacks the CarPUL pathway, and correspondingly, does not grow on or modify carrageenan.

There is precedent for our approach within the PUL literature. Mutagenesis is almost exclusively done using a Δtdk mutant lab strain of *Bacteroides thetaiotaomicron*. There is also a less used *Bacteroides uniformis* mutagenic type-strain available. We have personally used the *B. thetaiotaomicron* strain on many occasions to delete and insert genes under selection (10.1186/s40168-020-00975-x; 10.1038/s41598-019-53726-w; 10.1038/s41396-019-0406-z; 10.1038/s41564-017-0079-1; 10.1038/nature21725; 10.1038/nature13995). Importantly, however, these *B. thetaiotaomicron* and *B. uniformis* strains do not possess CarPULs, and therefore, cannot be used to specifically to address the reviewer's concern.

We feel we have provided very strong support for the role of the CarPUL in carrageenan utilization in the original manuscript using culturing and recombinant protein biochemistry, including x-ray crystallography, to demonstrate the enzymes from this pathway are active on carrageenan. Notably, there are many exemplary publications in the literature that characterize polysaccharide-PUL relationships without using mutagenesis. Here are a few examples:

1. Substrate: xyloglucan, Species: *Bacteroides ovatus*, DOI: 10.1098/rsob.160142
2. Substrate: mixed-linkage β -glucans, Species: *multiple Bacteroidota*, DOI: 10.1038/s41522-024-00578-6
3. Substrate: xylans, Species: unknown *Bacteroidaceae* MAG, DOI: 10.1038/s41467-022-28310
4. Substrate: arabinans, Species *Mesofavibacter profundus*, DOI: 10.1186/s40168-023-01618-7
5. Substrate: fucoidan, Species: *Planctomycetota* strains, DOI: 10.1038/s41467-024-55268-w
6. Substrate: arabinoxylans, Species: *Bacteroides intestinalis*, DOI: 10.1038/s41467-020-20737-5
7. Substrate: alpha-glucan, Species: *Polaribacter*, DOI: 10.1038/s41467-024-48301-5

In addition, a quantitative approach for selective carrageenan uptake has been incorporated (Ex Data Fig 3). This shows a discernible difference between the two strains (*BxCar5_{BOV}* and *BxCar17_{BOV}*). The methods are described on lines 937-956.

Q2.2. Another drawback of this study is that the association of *Bacteroides* with *M. japonicum* diet has been only shown in one cohort of cattle. The authors should discuss the limitations of this in the results or discussion section.

Response: We thank the reviewers for their concern. We understand that adequate replication of these findings is critical, which is why it should be noted that two distinct cohorts of cattle were used in this study. Two separate groups of cattle (Lines 635 - 647) were used in the *ad libitum* study and the silage studies, respectively. We have tried to make these distinctions more clear in the results.

Line 107: "Two cattle trials with distinct animals were conducted using *M. japonica* supplemented diets."

Q2.3. The title needs to be rephrased. Instead of latent traits – mention what traits; instead of genetic "dark matter" mention what genetic elements they have identified. Same goes with the

abstract as it vaguely mentions that they “characterized the gene to provide insights”. The authors should mention specifically (in general terminology) what characterization they have done and how it is novel.

Response: Thank you for the suggestion. We have clarified that the described latent traits refer to the “microbial carrageenan foraging pathways”, by inserting “their”.

In regards to “genetic dark matter” is a commonly used term to explain uncharacterized, hidden function encoded within genetic content. The latent traits (i.e. CarPULs) were discovered in the genetic dark matter. A PubMed search of “Dark Matter” within titles of published papers revealed an abundance of hits related to microbiome, genomes, and DNA. For example, a recent title from Nature (PMID: 40562910): “DeepMind’s” new Alpha Genome AI tackles the ‘dark matter’ in our DNA. Another example from Nature Communications (PMID: 37644066) “Interrogating the viral dark matter of the rumen ecosystem with a global virome database.” We feel this term is appropriate and fitting, considering we have discovered and characterized CarPULs within the dark matter of bovine microbiomes.

Previous Title: “Global distribution of microbial carrageenan foraging pathways reveals widespread latent traits within the “dark matter” of ruminant intestinal microbiomes

New Title: “Global distribution of microbial carrageenan foraging pathways reveals their widespread presence as latent traits within the “dark matter” of ruminant intestinal microbiomes”

To improve the abstract, we have replaced “enzymes” with “carrageenases” and stated how they were characterized: “Carrageenan-active polysaccharide utilization loci (CarPULs) were identified and recombinant GH16 subfamily 17 carrageenases were characterized informing novel substrate specificities for the subfamily, and providing insights into pathway specialization of divergent CarPULs”.

Q2.4. The authors provide limited details on the recombinant protein purification protocol, such as buffer compositions and elution gradients used. Could the authors include a more comprehensive description of these methods in the main text or supplementary materials to enhance reproducibility?

Response: The following Lines have been added to the methods:

Lines 798-800: “Protein was eluted by an increasing gradient 0–500 mM imidazole in 20 mM Tris pH 8.0 and 500 mM NaCl. Samples were buffer exchange with 20 mM Tris, 500 mM NaCl, and 2% glycerol using ultrafiltration column...”

Lines 849-851: “Washes were performed with binding buffer containing 10, 20, then 30 mM imidazole. Elution was performed with binding buffer containing 500 mM imidazole.”

Q2.5. Since part of the study involves characterization of the enzyme, could the authors provide an SDS-PAGE gel image of the purification, gel filtration chromatogram to determine the stoichiometry of the enzyme. It is important to confirm the purity of the purified protein(s) as well as their oligomeric state when used for assays, such as for BxMAGBOV GH16_17A or GH16_17B? This would strengthen the reliability of the enzymatic data.

Response: A gel of IMAC purification and the following chromatogram for the SEC run has been provided here for BxMAG_{BOV} GH16_17A. We note that the reproducible generation of diffraction quality crystals indicates a very high standard of purity. A brief analysis of the quaternary state in

the crystal lattice has been added to the main text; “The protein crystallized with a single monomer in the asymmetric unit. An analysis of interactions in the crystal lattice using PISA indicates a lack of stable quaternary structures.” Lines 190-191.

A

B

Figure Title: Purification of BxMAG_{Bov}A. **A)** SDS-PAGE gel of IMAC purification. The protein in the elution fraction was concentrated and further purified by size exclusion chromatography. **B)** Size exclusion chromatogram of protein from panel A. Though the protein had a slight tendency to aggregate the fractions chosen for analysis had an elution time consistent with a monomer.

The red box indicates fractions chosen to be concentrated and used for biochemical analysis and crystallization.

Q2.6. The authors demonstrate phylogenetic variability in GH16_17 genes across diverse species (Fig. 3A). Could the authors further investigate whether the active site architecture is conserved across these variants by modeling the active sites of representative GH16_17 enzymes to highlight key binding site residues?

Response: Thank you for the helpful suggestion. In response, we have completed structural alignments of GH16_17 members characterized in the literature, this study, and the GH16_13 member: WP_083194645.1; Cgbk16A_Wf (Extended Data Figure 4A). We have also completed a ConSurf analysis of BxMAG_{BOV} GH16_17A and included this result in Extended data figure 4B and Supplemental table 18.

A

B

Extended Data Figure 4: Sequence alignment of GH16_17 members. A) Sequence alignment of characterized GH16_17 members, and *BxCarPUL_{BoV}* GH16s. GH16_13 member

(WP_083194645.1; Cgbk16A_Wf) was added as a outgroup. Catalytic acid and nucleophiles glutamic acids (E) are denoted with black triangles. The black star denotes the amino acid, glycine – G or glutamic acid – E, which allows for or occludes 4S-modified anhydro-galactose in the -1 subsite, respectively. **B)** ConSurf analysis of BxMAG_{BOV} GH16_17A. 92 sequences were aligned with BxMAG_{BOV} using default settings with the color score present in the legend. Catalytic residues are labelled with the percent abundance of amino acid variety in that position. E280, the active site residue occluding or allowing for 4S-modified anhydro-galactose, is labelled. The ConSurf table is found in **Supplementary Table 18**.

Q2.7. The authors selected BxCar5BOV and BxCar17BOV for long-read genome sequencing (Lines 149–150) but focused biochemical characterization on BxMAGBOV GH16_17A and GH16_17B. Could the authors clarify why they were not shortlisted for characterization?

Response: The strategy of this work was to focus on novel structural motifs which may alter the specificity of the PULs. This hypothesis was proven true, as we were able to characterize a novel activity for GH16_17. Due to the intensive nature of biochemical characterization and crystallographic procedures, The first dataset created – BxMAG_{BOV} – was the source chosen for the GH16_17 enzymes. BxCar5_{BOV} PUL-1 and BxMAG_{BOV} contain near identical PULs, notably GH16_17A, GH16_17B are identical (100%; supplemental table 3) between these genomes. Therefore, it was not deemed critical to switch enzyme targets. Further, BxCAR17_{BOV} contains a single copy of GH16_17 identical to that characterized in Pudlo et al., 2022 (DOI: 10.1016/j.chom.2022.02.001) and was therefore, not a deemed a high interest target for characterization.

Q2.8. The E280 residue in BxMAGBov GH16_17A is critical for substrate specificity (Line 190). Could the authors confirm whether this residue is conserved in the corresponding position of GH16_17 enzymes from BxCar5BOV (CarPUL-1 or CarPUL-2) and BxCar17BOV, using sequence alignments or structural modeling?

Response: Please see Q2.6

Q2.9. In Fig. 3B–C, the structural figures display all active site residues as stick models, which may obscure key features. Could the authors simplify these panels by highlighting only the critical E280/G258 residue and the substrate to improve clarity and emphasize the structural determinant of substrate specificity?

Response: Fig 3B & C have been modified as requested. Stick models were removed, except for catalytic residues.

Q2.10. The authors note that BxCar17BOV has one CarPUL, while BxCar5BOV has two, with CarPUL-1 sharing 99–100% identity with BxMAGBOV's CarPUL (Lines 162–164). Could the authors provide the sequence identity between the GH16_17 enzyme in BxCar17BOV's CarPUL and BxMAGBOV's GH16_17 enzymes? Additionally, do the GH16_17 enzymes from BxCar17BOV, BxCar5BOV (CarPUL-1 and CarPUL-2), and BxMAGBOV share similar active site architectures, as determined by sequence alignment or structural modeling?

Response: Please see response to Q2.6. Further, as we understand there was issues with the attachment of supplementary tables, we would direct the reviewers to supplementary table 3 in which all blast hits between the bovine CarPULs can be observed. BxCAR17_{BOV} GH16_17 displayed the highest identity observed towards BxMAG_{BOV} GH16_17A (65%). Further,

structural alignments seen in extended data 4A (above) demonstrate that BxCAR17_{BOV} GH16_17 contains a glutamic acid (E) in the catalytic site.

Q2.11. Line 458, it should be kappa carrageen's, not "k"

Response: Corrected.

Q2.12. The data refinement statistics are missing the cell dimensions. Please incorporate that.

Response: The cell angles were omitted because in an orthorhombic space group they are necessarily all 90° and it is common practice to simplify the table in this case (as it is to provide only a b angle in monoclinic space groups, etc). Nevertheless, because this is supplementary material and space is not an issue, the angles have been added as requested.

Q2.13. Can the authors provide the PDB validation report for reviewing?

Response: This was attached with the initial submission. We understand that not all files were circulated to reviewers.

Reviewer #3 (Remarks to the Author):

Response: thank you for your contributions towards improving our manuscript.

Reviewer #1 (Remarks to the Author):

Q1.1: Including metagenomic sequencing and proteomics data for the *Mazzaella japonica* surface associated microbiome (MjSM) in main figures would allow for a thorough comparative analysis of its digestive fate and microbial effects and help assess its true impact on the ruminant microbiome.

A1.1: We thank the reviewer for their suggestion. To clarify, the *MjSM* is the microbiota scraped from the surface of *M. japonica* that was collected directly from the beach. The seaweed-associated microbial community is very different that the microbial community from the digestive organs of the animal. We have made a more detailed description of this in the methods:

Lines 738-742: “While environmental samples (soil, compost, and filter paper of *M. japonica*) were sequenced and analyzed using the same workflow, these results were omitted from downstream analysis and interpretation of data, as they were outside the primary scope and would confound the conclusions of this study. The data has been made available in the PRIDE repository.” This dataset was intended to provide an inventory of genes present on the marine environment microbiome to determine if candidates for horizontal gene transfer events from the seaweed to the bovine microbiome could be identified. Hence the inclusion of the *MjSM* microbiome was only included in the supplemental material as a negative result. We have changed the acronym to “*M. japonica* surface-associated microbiota (*MjSM*)” for clarity **(Line 267)**.

Q1.2: I thank the authors for their response, but I remain concerned that omitting the *Mazzaella japonica* surface-associated microbiome (*MjSM*) from the analyses and manuscript leaves a critical gap. In particular, the *Bacteroides* isolates and associated enzymatic functions reported in this study may in fact derive from the *MjSM* rather than being intrinsic to the rumen or faecal microbiome. Without a direct comparative analysis, it is difficult to exclude this possibility.

I therefore strongly encourage the authors to include at least a summary comparison between *MjSM* and the rumen/faecal datasets, highlighting whether or not *Bacteroides* species and associated carbohydrate-active enzymes overlap. This would not only address the concern but also provide transparency and accessibility of the results. Also the pride repository and figshare were not accessible for the reviewers.

A1.2: We have made the following changes to the manuscript to improve clarity, transparency and provide a summary of the *MjSM* data:

A1.2.1: The metaproteomic datasets presented within this study demonstrate that CarPUL genes within the *BxMAG* are not detected within the *MjSM* samples; however, as pointed out by the reviewer, this is not clearly addressed in the manuscript. Therefore, callouts for metaproteomic datasets (previously Supplementary Table 13 & 14) have been moved within the main text for more visibility (Supplementary Table 3 &

4; **Line 144**). We have also clarified in the text which MAG assembly is associated with *BxMAG_{BOV}* to avoid confusion (**Line 124**: “*BxMAG_{BOV}* (Assembly_18_bin.6)”).

Furthermore, the section describing the limited homology between genes from the ruminant-associated CarPULs and the *MjSM* has been edited to better reflect the metagenomic and metaproteomic findings:

Line 271-275: “The highest identity between ruminant-associated CarPULs and the *MjSM* was observed at the gene level between sulfatases (70-84% identity for sulfatase 1_30 vs. *MjSM*; Supplementary Table 110), and with identity above 50% for GH2, GH110, and GH167 homologs. Proteins for these genes were not detected in the *MjSM* metaproteome suggesting these genes were not expressed (Supplementary Tables 3 & 4).”

A1.2.2: The ruminant-sourced CarPUL genes were BLASTED against genes found on the *MjSM* in supplementary Table 11 (see Q1.1). These findings suggest that while homologous genes exist between ruminant-sourced CarPULs and the *MjSM*, the functional and evolutionary relationship between them is inconclusive.

A1.2.3: We have included the database of *MjSM* proteins into the Figshare to improve transparency, as requested, and made a note of this in the methods section (**Line 787**: “*MjSM* protein database”) and data availability statement (**Line: 985-986**).

A1.2.4: to address the point “...whether or not *Bacteroides* species and associated carbohydrate-active enzymes overlap”, we have improved the clarity of Supplementary Table 1 (Bovine MAG statistics and GTDB-tk taxonomy) by changing the title from “Bovine MAG statistics and GTDB-tk taxonomy.” to “Statistics and GTDB-tk taxonomy for MAGs generated in this study”. This table includes MAGs generated from *MjSM*.

Of note, no *Bacteriaceae* family genomes were identified within the *MjSM*, which is consistent with the literature on marine *Bacteroidota* (10.1038/ismej.2012.169; <https://doi.org/10.1038/s41598-024-69362-y>).

A1.2.5: We have improved upon the discussion section of the text to better summarize these findings:

Line 399-401: “Although homologs of ruminant-associated CarPUL genes were found within marine-sourced sequence datasets and the *MjSM*, there was not convincing evidence to conclude HGT from these microorganisms (Supplementary Table 11).”

A1.2.6: In addition to these edited changes in the manuscript, we contend that the presence of CarPULs found in geographically distributed host species, as determined by metagenomic studies and genomes from bacterial isolates, demonstrates that the CarPULs studied here do not originate from the *MjSM* or a recent HGT event.

We apologize for oversight on the availability of the PRIDE and FigShare database. Please see the below reviewer access tokens:

PRIDE database:

Username: reviewer_pxd060679@ebi.ac.uk

Password: IOfr3A2xl8tk

Figshare:

<https://figshare.com/s/79334adffee438e831a0>

<https://figshare.com/s/b29b628f451e1c4b92e0>

Reviewer #2 (Remarks to the Author):

Q2.1: While the authors correctly acknowledge that generating knockouts in environmental *Bacteroides* isolates is technically challenging, the current evidence remains strictly correlative rather than causative. Without genetic validation, the functional link between the CarPUL and carrageenan metabolism cannot be definitively established.

A gain-of-function experiment would provide the most direct and interpretable validation. For example, introducing the CarPUL locus (e.g., from BxMAGBOV) into a genetically tractable host that lacks the native pathway, such as *Escherichia coli* or a laboratory *Bacteroides* strain, and subsequently testing for the ability to degrade *M. japonica* extracts using established assays (e.g., FACE or LC-MS; see Lines 201–211), would demonstrate sufficiency of the CarPUL for carrageenan utilization. Even a partial reconstruction or expression of key CarPUL enzymes in such hosts would meaningfully strengthen the mechanistic argument.

In addition to traditional mutagenesis, recent advances in *Bacteroides* genetics now enable CRISPR interference (CRISPRi)-based knockdown of individual genes without requiring double recombination or strain engineering. CRISPRi systems employing catalytically inactive Cas9 (dCas9) have been successfully adapted for *Bacteroides* (<https://pmc.ncbi.nlm.nih.gov/articles/PMC10861873/>). Even if environmental isolates are difficult to transform, such tools could be implemented in a CRISPRi-compatible model strain background to verify the contribution of specific CarPUL genes to carrageenan degradation. Attempting such an experiment, or at least discussing why it could not be done, would demonstrate due diligence.

If genetic perturbation truly remains infeasible (after attempting the experiment), the authors should explicitly acknowledge this limitation in the Discussion, stating that the findings establish associative evidence but stop short of demonstrating causality. Merely correlating the presence of CarPULs with carrageenan utilization, without any perturbation-based validation (knockout, knockdown, or complementation), does not meet the evidence required to infer functionality.

A2.1: We believe that structural and functional characterization of GH16_17 (Figure 3), growth profiles of *Bacteroides* spp. isolates with and without exogenous GH16_17B

(Figure 2, Extended Data Figure 5F), metaproteomic analysis of cattle-associated microbiomes fed *M. japonica* (Figure 1; Supplementary Table), alongside previously published work which confirmed the growth and activity of *Bacteroides* CarPULs (10.1016/j.chom.2022.02.001), Suggests that it is a conservative conclusion that CarPULs are strongly associated with carrageenan utilization in *Bacteroides*. We feel that this work is highly aligned, if not exceeds what is normally conducted for metagenomic PUL analysis and complements the microbial ecological implications drawn from this study. Nor do we believe that we overstep and conclusively state that CarPUL structures are the sole mechanism of carrageenan depolymerization anywhere in the manuscript.

To ensure this is clearly articulated and in response to the editors comment to soften the claims made within the manuscript, we have made the following edits:

Line 314-321: “Although there is potential for other enzymes to be involved in carrageenan utilization, CarPULs appear to contain the catalytic machinery required for carrageenan polymerization⁵². No other CAZymes belonging to known carrageenan-specific families were identified in the bovine-associated microbial genomes and metagenomes, or upregulated within the metaproteomic datasets (Supplementary Table 4). As carrageenan depolymerization requires a consort of CAZymes, sulfatases, and metabolic genes which is reflected in the structural heterogeneity of CarPUL structures²⁰ (Fig. 4B), these results suggest that CarPULs encode the conserved core machinery for carrageenan utilization within *Bacteroides* species.”

In addition we would like to draw attention to other instances in the manuscript where we leave room for variances that may exist between CarPUL function and carrageenan utilization:

Line 221-223: “Treating *MjEx* with BxMAG_{BOV} GH16_17B also improved the growth of the BxCar17_{BOV} isolate, which lacks a homologous enzyme (Extended Data Fig. 5F), suggesting it has a biological role as a keystone enzyme in κ/I-carrageenan utilization.”

Line 295-298: “Metagenome-guided metaproteomics analysis presented here suggests that *Bacteroides*-affiliated MAGs are one of the primary mechanisms for microbial degradation and utilization of *M. japonica* carrageenans in the lower GIT of cattle (Fig. 1; Extended Data Fig. 2C).”

Reviewer #3 (Remarks to the Author):

No response necessary.